# WRF4CIR: Weight-Regularized Fine-Tuning Network for Composed Image Retrieval

## Abstract

Composed Image Retrieval (CIR) task aims to retrieve target images based on reference images and modification texts. Current CIR methods primarily rely on fine-tuning vision-language pre-trained models. However, due to the large-scale nature of pre-trained models and the limited training data for CIR task, significant overfitting commonly occurs during fine-tuning, resulting in poor generalization. To address this issue, we introduce WRF4CIR, a Weight-Regularized Fine-tuning network for CIR. Specifically, during the fine-tuning process, we apply adversarial perturbations to the model weights for regularization, where these perturbations are generated in the opposite direction of gradient descent. Intuitively, WRF4CIR increases the model's learning difficulty on training data, effectively mitigating overfitting. Technically, WRF4CIR explicitly regularizes the flatness of the weight loss landscape, enhancing the model's robustness to weight perturbations and improving generalization. Extensive experiments on benchmark datasets demonstrate that WRF4CIR significantly narrows the generalization gap and achieves substantial improvements over existing methods.

## 1 Introduction

Composed image retrieval (CIR) is a task that retrieves target images based on reference images and modification texts. By integrating visual and textual information, the CIR model enables more accurate and flexible retrieval than traditional retrieval methods, showing great potential in real-world applications like e-commerce (Goenka et al., 2022; Wu et al., 2021), medical diagnostics (Cao et al., 2014), and document search (Hassan et al., 2013). The CIR framework typically involves two key components: feature encoder and multimodal fusion module. In general, the reference image and text are first processed by feature encoders, then their features are fused into a unified representation to retrieve the target image.

The feature encoder in previous CIR methods (Vo et al., 2019; Chawla et al., 2021; Delmas et al., 2022) primarily relied on traditional models, such as ResNet (He et al., 2016) for image and LSTM (Hochreiter and Schmidhuber, 1997b) for text. Due to the limited multimodal feature encoding capabilities of traditional models, these methods can only achieve suboptimal performance. To overcome this limitation, recent research (Baldrati et al., 2023; Bai et al., 2024; Tian et al., 2025) employ vision-language pre-trained models (VLPs) as feature encoder. They leverage the powerful feature extraction and cross-modal alignment capabilities of VLPs, yielding significant improvements in CIR benchmark. However, due to the large-scale nature of pre-trained models and the relatively scarce training data for CIR task, significant overfitting commonly occurs during fine-tuning, resulting in poor generalization.

In this work, we investigate the overfitting problem in VLP-based CIR task. First, we argue that overfitting is a prevalent phenomenon in VLP-based CIR models. To substantiate this, we fine-tune different pre-trained models and visualize the learning curves across a range of CIR methods and datasets. We observe a progressively widening gap in recall between the training and test sets, which eventually manifests as a significant generalization gap under all experimental settings. These results suggest that overfitting is a general and widespread issue in VLP-based CIR task. To further analyze the overfitting phenomenon, we partition the CIR datasets into different proportions and fine-tune pre-trained models on subsets of varying sizes. We observe a clear correlation between the size of the training data and the generalization gap: the smaller the training set, the larger the gap.

The experimental results indicate that the limited amount of training data in CIR contributes to the overfitting observed during fine-tuning of pre-trained models. However, annotating data for CIR task is both time-consuming and labor-intensive, which raises a key challenge:

*how can we effectively fine-tune pre-trained models with limited CIR training data?*

To address this issue, motivated by prior work (Wu et al., 2020; Foret et al., 2021) that mitigates overfitting by flattening the weight loss landscape, we introduce a Weight-Regularized Fine-tuning network for Composed Image Retrieval, termed WRF4CIR. It regularizes the fine-tuning process by introducing adversarial perturbations to the model weights. Specifically, in a minibatch, we apply an adversarial perturbation to the model weights in the direction opposite to gradient descent, and remove it after the parameter update to maintain training stability. Intuitively, these adversarial perturbations increase the model's learning difficulty on the training data, thus can effectively mitigate overfitting. Technically, WRF4CIR explicitly regularizes the flatness of weight loss landscape, which helps narrow the generalization gap and improve model performance. The effectiveness of the proposed method is validated across different pre-trained backbones, fusion strategies, and benchmark datasets. Extensive experiments demonstrate that our method achieves substantial improvements over existing methods without relying on large language models or additional training data. In summary, our main contributions are as follows:

- We find that overfitting is a prevalent phenomenon in VLP-based CIR task, consistently appearing across different methods and datasets.
- We introduce a weight-regularized fine-tuning network for composed image retrieval, which mitigates overfitting and improves model performance by explicitly flattening the weight loss landscape.
- Extensive experiments demonstrate that our method achieves substantial improvements over existing approaches on the FashionIQ and CIRR datasets.

## 2 RELATED WORK

### 2.1 COMPOSED IMAGE RETRIEVAL

Composed image retrieval (CIR) aims to retrieve a target image by jointly leveraging a reference image and a textual modification. Previous approaches (Vo et al., 2019; Chawla et al., 2021; Delmas et al., 2022; Kim et al., 2021; Lee et al., 2021; Yang et al., 2021) mainly utilized conventional networks (e.g., ResNet (He et al., 2016) or LSTM (Hochreiter and Schmidhuber, 1997b)) as feature encoders, but their performance is suboptimal due to limited multimodal feature encoding capability. Recent methods (Baldrati et al., 2023; Bai et al., 2024; Chen et al., 2024; Wen et al., 2023; Jiang et al., 2024; Tian et al., 2025) have overcome this limitation by leveraging the powerful representation capabilities of vision-language pre-training models (VLPs), achieving significantly improved results. For example, CLIP4CIR (Baldrati et al., 2023) is the first to employ CLIP (Radford et al., 2021) for extracting image and text features, which are then integrated via a gating mechanism. Bai et al. (2024) use the Q-former from BLIP-2 (Li et al., 2023) and enable fusion at the input level by projecting images into the word embedding space of the VLPs. Despite the significant performance gains achieved by VLP-based CIR methods, they commonly suffer from severe overfitting during the fine-tuning of pre-trained models. Several existing methods (Feng et al., 2024; Ge et al., 2025) have leveraged large language models (LLMs) for data augmentation, which has been observed to improve generalization. For example, SPN4CIR (Feng et al., 2024) leverage LLMs to generate additional triplets for positive sample augmentation. In contrast to these approaches that rely on extra data, our work focuses on the fine-tuning strategy for pre-trained models on relatively limited CIR data, aiming to mitigate overfitting through weight regularization and improve retrieval performance.

### 2.2 VISION-LANGUAGE PRE-TRAINING MODELS

Vision-language pre-training models trained on large-scale datasets can effectively align visual and textual information. Various model architectures (Jia et al., 2021; Wang et al., 2021; 2023; Li et al., 2022) and pre-training objectives (Radford et al., 2021; Li et al., 2021; 2023) have been introduced over time, continuously enhancing multimodal encoding capabilities and generalization on diverse

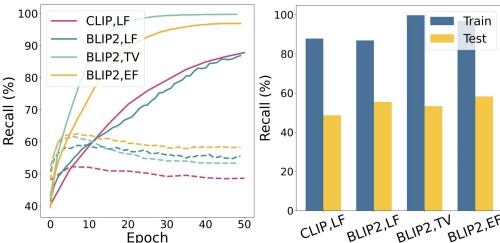 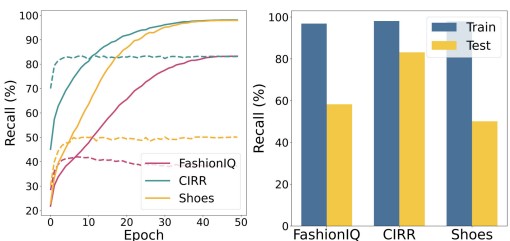

(a) Across different baseline methods (Solid line: training set; Dashed line: val set)

(b) Across different datasets (Solid line: training set; Dashed line: val set)

Figure 1: Visualization of learning curve and generalization gap across (a) different baseline methods on FashionIQ (LF: late-fusion; EF: early-fusion; TV: sentence-level text inversion); and (b) different datasets, including FashionIQ, CIRR and Shoes.

downstream tasks (Song et al., 2022; Sun et al., 2024a; Mokady et al., 2021). Recently, CLIP (Radford et al., 2021), BLIP (Li et al., 2022) and BLIP-2 (Li et al., 2023) have been widely used in CIR methods. In this work, we adopt these pre-trained models as backbones to investigate the overfitting problem in VLP-based CIR task.

### 2.3 OVERFITTING AND REGULARIZATION

Deep models excel at learning the rich representations needed for complex tasks but tend to overfit on small datasets (Dinh et al., 2017; Zhang et al., 2016; Kawaguchi et al., 2017; Olson et al., 2018; Power et al., 2022). Recently, several studies (Dinh et al., 2017; Izmailov et al., 2018; Li et al., 2018; Keskar et al., 2016) have explored the connection between the flatness of local minima and generalization ability. Hochreiter and Schmidhuber (1994; 1997a) first revealed that flatter minima in the loss landscape are associated with better generalization performance. Keskar et al. (2016) and Dinh et al. (2017) used the eigenvalues of the Hessian to theoretically explain this phenomenon, while Jiang et al. (2019) further validated this insight under various settings and hyperparameters. Recent work (Wu et al., 2020; Foret et al., 2021; Kwon et al., 2021; Wen et al., 2022) leverages this connection to improve generalization by flattening the weight loss landscape. For example, Wu et al. (2020) reduce the robust generalization gap by incorporating adversarial weight perturbation (AWP) into adversarial training. Motivated by these findings, we further investigate the overfitting phenomenon in VLP-based CIR from the perspective of the weight loss landscape. Unlike prior work, our study focuses on overfitting in VLP-based CIR, a multimodal retrieval task involving the fine-tuning of large vision-language models on triplet data. To the best of our knowledge, our work is the first to reveal that overfitting is a prevalent phenomena in VLP-based CIR, and we demonstrate that our method effectively alleviates this issue.

## 3 OVERFITTING ISSUE IN VLP-BASED COMPOSED IMAGE RETRIEVAL

In this section, we first visualize the overfitting phenomenon in VLP-based composed image retrieval. We then provide a detailed analysis of this overfitting issue, and finally introduce the motivation behind our proposed method.

**Visualization.** In VLP-based CIR, the fine-tuning process is typically monitored using validation recall, potentially overlooking the model's behavior on the training set. To more comprehensively characterize the fine-tuning process, we record and report the model's recall on both the training and validation sets, which better reflects the model's generalization ability. Specifically, we conduct experiments using widely adopted VLP backbones and fusion strategies from recent state-of-the-art baselines (Baldrati et al., 2023; Jiang et al., 2024; Tian et al., 2025), with implementation details provided in Appendix A.1. We visualize the learning curves and generalization gaps across different baseline methods and datasets (FashionIQ (Wu et al., 2021), CIRR (Liu et al., 2021), and Shoes (Guo et al., 2018)), and the results are presented in Figure 1. As illustrated in Figure 1(a), most methods achieve over 90% average recall on the FashionIQ training set upon convergence, while their average recall on the validation set plateaus at around 50%. Notably, the best validation performance typically occurs within the first five epochs—well before the model fully converges on the training

set. Further training continues to improve training recall but yields no gain on the validation set, indicating clear signs of overfitting. Similar phenomena are observed across different datasets, as shown in Figure 1(b). Overall, we consistently observe a significant generalization gap under all experimental settings, indicating that overfitting is a prevalent phenomenon in VLP-based CIR task.

**Analysis about the overfitting issue.** To further analyze this overfitting phenomenon, we fine-tune pre-trained models with varying sizes of training data. Specifically, we randomly split the training sets into subsets of different sizes with proportions {0.2, 0.4, 0.6, 0.8, 1.0} and conduct experiments under three datasets (FashionIQ, CIRR, and Shoes) and two pre-trained models (CLIP and BLIP-2). The generalization gap and validation recall for BLIP-2 on the FashionIQ dataset is shown in Figure 2, and detailed results for each dataset and additional model are provided in Appendix A.2. It is observed that as the size of the training data decreases, the validation recall drops and the generalization gap continues to widen. These experimental results indicate that the limited amount of training data contributes to the overfitting phenomenon during the fine-tuning of pre-trained models. However, annotating data for CIR task is both time-consuming and labor-intensive. Moreover, CIR data generated by large language models often suffer from inconsistent quality (Feng et al., 2024; Levy et al., 2024; Zhou et al., 2025). This naturally raises the question of how to efficiently fine-tune pre-trained models with limited CIR training data.

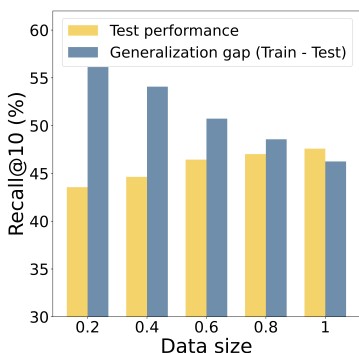

Figure 2: Generalization gap and recall of BLIP-2 under varying training data sizes.

**Motivation of WRF4CIR.** To address this issue, a common approach is to employ regularization techniques, since they can effectively reduce the risk of overfitting caused by limited training data and excessive model complexity (Geman et al., 1992; Belkin et al., 2019). Furthermore, inspired by previous studies (Wu et al., 2020; Foret et al., 2021) that improve generalization by flattening the weight loss landscape, we introduce a weight-regularized fine-tuning network for CIR, named WRF4CIR, which regularizes the fine-tuning process by applying perturbations to the model weights. The optimization objective of WRF4CIR is formulated as follows:

$$\min_{\theta}\{\mathcal{L}(\theta) + (\mathcal{L}(\theta + \delta) - \mathcal{L}(\theta))\} \rightarrow \min_{\theta} \mathcal{L}(\theta + \delta). \tag{1}$$

Here, $\mathcal{L}(\theta)$ denotes the standard optimization objective of the model, which, in the context of CIR, corresponds to the contrastive loss between the target representation and the query representation. The term $(\mathcal{L}(\theta + \delta) - \mathcal{L}(\theta))$ serves as a regularization component that reflects the flatness of the weight loss landscape, where $\delta$ represents a perturbation applied to the model weights. In the next section, we introduce the implementation details of WRF4CIR.

# 4 WEIGHT-REGULARIZED FINE-TUNING NETWORK FOR COMPOSED IMAGE RETRIEVAL

**Basic framework.** Composed image retrieval (CIR) aims to retrieve target images $I_t$ from a large-scale database using multimodal queries $\{I_r, T\}$. $I_r$ and $T$ are processed by given feature encoders, after which their features are fused into a unified representation to retrieve the target image $I_t$. In this part, we first present a basic CIR framework based on BLIP-2, which consists of an image encoder and a lightweight Q-Former, as shown in Figure 3. Specifically, the framework employs a frozen image encoder to extract initial features from both the reference image $I_r$ and the target image $I_t$. These features are then passed to the Q-Former, where they interact with learnable query embeddings $q$ and tokenized text, producing fused query representations $u$ and target image representations $v$. Finally, the fused query representation $u$ and the target representation $v$ are aligned by optimizing a contrastive loss, which aims to minimize the distance between positive pairs while maximizing it for negative pairs. The contrastive objective is formulated as:

$$\ell_{q2t}(\hat{u}, \hat{v}) = -\frac{1}{|\mathcal{B}|} \sum_{i \in \mathcal{B}} \log \frac{\exp\left(\tau \hat{u}_i^T \hat{v}_i\right)}{\sum_{j \in \mathcal{B}} \exp\left(\tau \hat{u}_i^T \hat{v}_j\right)}. \tag{2}$$

Here, $\hat{u}$ and $\hat{v}$ are the normalized vectors of $u$ and $v$, $\mathcal{B}$ is a randomly sampled mini-batch, and $\tau$ is a temperature coefficient for scaling in contrastive loss.

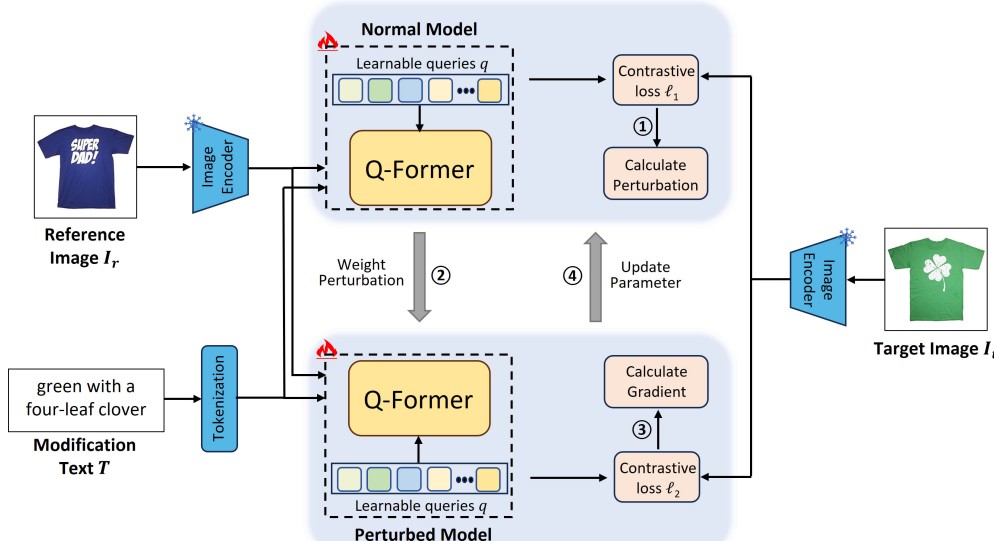

Figure 3: **Illustration of our WRF4CIR framework.** The reference image and modification text are first fused through the Q-Former, then aligned with the target image by optimizing a contrastive loss. Building on this process, adversarial perturbations are generated and applied to the model weights for regularization. Finally, the gradients computed from the perturbed model are used to update the parameters of the normal model.

**Weight perturbation strategy.** The proposed WRF4CIR introduces perturbations to the model weights in order to regularize the weight loss landscape and mitigate overfitting. The optimization objective of WRF4CIR has been defined in Equation (1). Here, we mainly discuss the choice of perturbation strategy in WRF4CIR. Although random weight perturbation offers a simple implementation, its arbitrary direction may interfere with model optimization and lead to suboptimal performance. To overcome this limitation, we adopt a gradient-based adversarial perturbation strategy, similar to AWP (Wu et al., 2020), where perturbations are applied to the model weights in the direction opposite to their gradients. Intuitively, these adversarial perturbations increase the learning difficulty on the training data, thereby effectively alleviating overfitting. From a technical perspective, the proposed method explicitly regularizes the flatness of the weight loss landscape, enhancing the model's robustness to weight perturbations and improving its generalization ability. Thus, the optimization objective of WRF4CIR can be further expressed as follows:

$$\min_{\theta} \max_{\delta} \mathcal{L}(\theta + \delta) \rightarrow \min_{\theta} \max_{\delta} \ell_{q2t}\left(f_{\theta+\delta}\left(I_r, T\right), f_{\theta+\delta}\left(I_t\right)\right) \quad \text{s.t.} \quad \|\delta_l\| \leq \gamma \|\theta_l\|. \quad (3)$$

Here, $\theta$ denotes the model weights, $\delta$ denotes the weight perturbation, and $\theta_l$ ($\delta_l$) denotes the model weights (weight perturbations) in the $l$-th layer. The perturbed model is denoted by $f_{\theta+\delta}$. For the perturbation constraint, we follow the setup adopted in the adversarial training scenario (Wu et al., 2020), where the magnitude of the weight perturbation is kept proportional to the norm of the model weights in each network layer. Meanwhile, a hyperparameter $\gamma$ is introduced to flexibly control the overall perturbation strength.

**Weight-regularized fine-tuning.** Building upon the basic CIR framework and the weight perturbation strategy, the overall pipeline of WRF4CIR is illustrated in Figure 3. Specifically, WRF4CIR performs two forward-backward propagation per mini-batch during training: one for the normal model and one for the perturbed model. For the normal model, we follow the process illustrated in the basic framework and compute the model gradients to determine the direction of weight perturbations. Based on the model gradients, adversarial perturbations $\delta$ are generated and constrained according to the norm of model weights and the predefined perturbation strength $\gamma$, as shown in the following equation:

$$\delta \leftarrow \gamma \frac{\nabla_{\theta}\ell_{q2t}\left(f_{\theta}\left(I_r, T\right), f_{\theta}\left(I_t\right)\right)}{\|\nabla_{\theta}\ell_{q2t}\left(f_{\theta}\left(I_r, T\right), f_{\theta}\left(I_t\right)\right)\|}\|\theta\|. \quad (4)$$

The weight perturbation $\delta$ is then added to the normal model to obtain the perturbed model, denoted as $f_{\theta+\delta}$. Finally, we optimize the perturbed model and use its gradients to update the parameters

of the normal model, which is equivalent to removing the weight perturbations after optimizing the perturbed model, as illustrated in the following equation:

$$\theta \leftarrow (\theta + \delta) - \eta \nabla_{\theta+\delta} \ell_{q2t} \left( f_{\theta+\delta} \left( I_r, T \right), f_{\theta+\delta} \left( I_t \right) \right) - \delta. \tag{5}$$

The complete procedure of WRF4CIR is summarized in Algorithm 1, where the weight regularization module is portable and can be seamlessly integrated into different VLP models and fusion strategies.

---

**Algorithm 1** WRF4CIR

---

1: **Input:** Training set $D_t = \{(I_r^i, T^i, I_t^i)\}_{i=1}^n$; perturbation strength $\gamma$; learning rate $\eta$ ; normal model $f$.
2: **Output:** Fine-tuned model $f$.
3: **for** $t = 0$ to $T - 1$ **do**
4:     Forward propagation and calculate gradients of normal model:
5:         $L(f) \leftarrow \nabla_{f_\theta} \ell_{q2t}(f_\theta(t; T, I_r), f_\theta(t; I_t))$
6:     Add adversarial perturbation: $f_{\theta+\delta}(t) \leftarrow f_\theta(t) + \gamma \frac{L(f)}{\|L(f)\|} \|f_\theta(t)\|$
7:     Forward propagation and calculate gradients of perturbed model:
8:         $L'(f) \leftarrow \nabla_{f_{\theta+\delta}} \ell_{q2t}(f_{\theta+\delta}(t; T, I_r), f_{\theta+\delta}(t; I_t))$
9:     Update parameter of normal model: $f_\theta(t + 1) \leftarrow f_\theta(t) - \eta L'(f)$
10: **end for**

---

## 5 EXPERIMENTS

### 5.1 EXPERIMENTAL SETUP

**Implementation details.** Our method is implemented based on PyTorch and runs on a single NVIDIA RTX A100 GPU with 40GB of memory. For a fair comparison, we adopt BLIP-2 ViT-G as the backbone, which has been widely adopted in recent state-of-the-art methods (Bai et al., 2024; Sun et al., 2024b; Tian et al., 2025; Li et al., 2025). Following the same setup as BLIP-2 based CIR methods (Bai et al., 2024), we initialize the image encoder (kept frozen during training) and treat the Q-Former along with learnable queries as trainable parameters. We use the AdamW (Loshchilov and Hutter, 2019) optimizer with a weight decay of 0.05. The input images are resized to 224 × 224, with a padding ratio of 1.25 to ensure uniformity. The learning rate is initialized to 1e-5 following a cosine schedule for both the **CIRR** (Liu et al., 2021) and **FashionIQ** (Wu et al., 2021) datasets. Before applying weight-regularized fine-tuning, we perform a 3-epoch warm-up phase, considering that the model does not exhibit overfitting during the initial training epochs.

**Datasets and metrics.** We evaluate our method on two CIR benchmarks: **FashionIQ** (Wu et al., 2021) contains fashion items of three categories: Dress, Shirt, and TopTee. Every triplet consists of a reference image, two relative captions and a target image. We evaluate our model on the validation set, which consists of 6K triplets. Following previous studies (Baldrati et al., 2023; Wen et al., 2023; Bai et al., 2024; Tian et al., 2025), we use the recall at rank $K$ (R@$K$) as the evaluation metric and calculate the average to assess the overall performance. **CIRR** (Liu et al., 2021) comprises almost 21K real-life open-domain images taken from the NLVR$^2$ dataset (Suhr et al., 2018). We assess our model on the test set of CIRR, which contains 4.1K testing triplets. We use the recall at rank $K$ where $K = 1, 5, 10, 50$ and Recallsubset@K as evaluation metrics. The final score is computed as the average of Recall@5 and Recallsubset@1.

### 5.2 PERFORMANCE COMPARISON

The experimental results on the FashionIQ dataset, evaluated on the original split of val set, are reported in Table 1. It can be observed that our method achieves the highest recall across most evaluation metrics on FashionIQ, except for Dress R@10, where it ranks second. Compared to VLP-based methods such as SPRC (Bai et al., 2024) and CCIN (Tian et al., 2025), our approach outperforms the best of them (SPRC) by 2.43% and 2.36% in average R@10 and R@50, respectively. These results demonstrate the effectiveness of the proposed method in improving model generalization and enhancing test performance. In addition, we also compare our method with approaches that leverage additional training data generated by large language models (LLMs). For instance, SPN4CIR

Table 1: **Results on FashionIQ val set.** The best and second-best results are marked in Red and Blue, respectively. The symbol † indicates methods that leverage additional data.

| Method | Backbone | Dress | | Shirt | | TopTee | | Average | | |
|---|---|---|---|---|---|---|---|---|---|---|
| | | R10 | R50 | R10 | R50 | R10 | R50 | R10 | R50 | Rmean |
| CLIP4CIR (Baldrati et al., 2023) | CLIP ResNet-50×4 | 38.18 | 62.17 | 44.01 | 64.57 | 45.39 | 69.56 | 42.52 | 65.60 | 54.06 |
| BLIP4CIR+Bi (Liu et al., 2024) | BLIP ViT-B | 42.09 | 67.33 | 41.76 | 64.28 | 46.61 | 70.32 | 43.49 | 67.31 | 55.40 |
| CASE (Levy et al., 2024) | BLIP ViT-B | 48.48 | 70.23 | 47.44 | 69.36 | 50.18 | 72.24 | 48.79 | 70.68 | 59.74 |
| CaLa (Jiang et al., 2024) | BLIP-2 ViT-L | 42.38 | 66.08 | 46.76 | 68.16 | 50.93 | 73.42 | 46.69 | 69.22 | 57.95 |
| Re-ranking (Liu et al., 2023) | BLIP ViT-B | 48.14 | 71.43 | 50.15 | 71.25 | 55.23 | 76.80 | 51.17 | 73.13 | 62.15 |
| SPRC (Bai et al., 2024) | BLIP-2 ViT-G | 49.18 | 72.43 | 55.64 | 73.89 | 59.35 | 78.58 | 54.92 | 74.97 | 64.85 |
| CIR-LVLM (Sun et al., 2024b) | Qwen-7B | 50.42 | 73.57 | 58.59 | 75.86 | 59.61 | 78.99 | 56.21 | 76.14 | 66.17 |
| CCIN (Tian et al., 2025) | BLIP-2 ViT-G | 49.38 | 72.58 | 55.93 | 74.14 | 57.93 | 77.56 | 54.41 | 74.76 | 64.59 |
| TME (Li et al., 2025) | BLIP-2 ViT-G | 49.73 | 71.69 | 56.43 | 74.44 | 59.31 | 78.94 | 55.15 | 75.02 | 65.09 |
| **WRF4CIR (ours)** | BLIP-2 ViT-G | 51.71 | 75.41 | 58.94 | 76.21 | 61.40 | 80.37 | 57.35 | 77.33 | 67.34 |
| DQU-CIR† (Wen et al., 2024) | CLIP ViT-H | 51.9 | 74.37 | 53.57 | 73.21 | 58.48 | 79.23 | 54.65 | 75.60 | 65.12 |
| SPRC+SPN† (Feng et al., 2024) | BLIP-2 ViT-G | 50.57 | 74.12 | 57.70 | 75.27 | 60.84 | 79.96 | 56.37 | 76.45 | 66.41 |

Table 2: **Results on CIRR test set.** The best and second-best results are marked in Red and Blue, respectively. The symbol † indicates methods that leverage additional data.

| Method | Backbone | Recall@K | | | | Recall$_{subset}$@K | | | Avg. |
|---|---|---|---|---|---|---|---|---|---|
| | | K=1 | K=5 | K=10 | K=50 | K=1 | K=2 | K=3 | |
| CLIP4CIR (Baldrati et al., 2023) | CLIP ResNet-50×4 | 38.53 | 69.98 | 81.86 | 95.93 | 68.19 | 85.64 | 94.17 | 69.09 |
| BLIP4CIR+Bi (Liu et al., 2024) | BLIP ViT-B | 40.15 | 73.08 | 83.88 | 96.27 | 72.10 | 88.27 | 95.93 | 72.59 |
| CASE (Levy et al., 2024) | BLIP ViT-B | 48.00 | 79.11 | 87.25 | 97.57 | 75.88 | 90.58 | 96.00 | 77.50 |
| CaLa (Jiang et al., 2024) | BLIP-2 ViT-L | 49.11 | 81.21 | 89.59 | 98.00 | 76.27 | 91.04 | 96.46 | 78.74 |
| TG-CIR (Wen et al., 2023) | CLIP ViT-B | 45.25 | 78.29 | 87.16 | 97.30 | 72.84 | 89.25 | 95.13 | 75.57 |
| Re-ranking (Liu et al., 2023) | BLIP ViT-B | 50.55 | 81.75 | 89.78 | 97.18 | 80.04 | 91.90 | 96.58 | 80.90 |
| SPRC (Bai et al., 2024) | BLIP-2 ViT-G | 51.96 | 82.12 | 89.74 | 97.69 | 80.65 | 92.31 | 96.60 | 81.39 |
| CIR-LVLM (Sun et al., 2024b) | Qwen-7B | 53.64 | 83.76 | 90.60 | 97.93 | 79.12 | 92.33 | 96.67 | 81.44 |
| CCIN (Tian et al., 2025) | BLIP-2 ViT-G | 53.41 | 84.05 | 91.17 | 98.00 | - | - | - | - |
| TME (Li et al., 2025) | BLIP-2 ViT-G | 53.42 | 82.99 | 90.24 | 98.15 | 81.04 | 92.58 | 96.94 | 82.01 |
| **WRF4CIR(ours)** | BLIP-2 ViT-G | 56.58 | 85.45 | 92.24 | 98.68 | 80.48 | 92.60 | 96.84 | 82.96 |
| DQU-CIR† (Wen et al., 2024) | CLIP ViT-H | 46.22 | 78.17 | 87.64 | 97.81 | 70.92 | 87.69 | 94.68 | 80.44 |
| SPRC+SPN† (Feng et al., 2024) | BLIP-2 ViT-G | 55.06 | 83.83 | 90.87 | 98.29 | 81.54 | 92.65 | 97.04 | 82.69 |

(Feng et al., 2024) generates 96,000 triplets as positive samples and reports the best performance gains with 12,000 extra triplets. In comparison, our method still achieves a 0.93% improvement in average recall without using any additional data. This highlights the efficiency of our approach in fine-tuning pre-trained models with limited CIR training data.

We further report results on the CIRR dataset. Note that this dataset is more challenging than FashionIQ, as the degree of overfitting and the generalization gap are much smaller compared to those in FashionIQ. As shown in Table 2, the proposed approach still outperforms all other methods in terms of average recall. Compared to recent methods that incorporate LLMs, our method can achieve better performance. For example, CIR-LVLM (Sun et al., 2024b) uses a LLM (Qwen-7B (Bai et al., 2023)) as a user intent-aware encoder, yet WRF4CIR yields a 1.52% improvement in average recall even without relying on LLMs. The superior performance of our method on this dataset further highlights the importance of leveraging the weight-regularized fine-tuning strategy for VLP-based CIR. In addition, we also provide visualization retrieval results in Appendix A.9 for qualitative analysis.

## 5.3 ABLATION STUDIES

In this subsection, we first analyze the effects of weight perturbation and present visualization results to evaluate the impact of our method on mitigating overfitting in CIR. We then assess its effectiveness across different CIR mechanisms, training data sizes, and LoRA ranks. Finally, we compare our method with other regularization strategies and provide an analysis of its computational cost.

**Analysis on weight perturbation.** We first investigate the effects of both the strength and direction of weight perturbations on model performance. The *perturbation strength* $\gamma$ (PS), defined as the ratio between the norm of weight perturbations and that of model parameters, determines the overall intensity of regularization applied to the model. To investigate its impact on retrieval performance, we conducted experiments under different perturbation strengths. As shown in Figure 4(a), increasing $\gamma$ from zero gradually improves performance on both CIRR and FashionIQ val sets. Experimental results show that WRF4CIR achieves superior performance on both the FashionIQ and CIRR datasets

Table 3: **Ablation studies** towards the weight perturbation strategy of WRF4CIR.

| Method | PD | PS | FashionIQ | | | CIRR | | | |
|---|---|---|---|---|---|---|---|---|---|
| | | | R10 | R50 | Rmean | R1 | R5 | R10 | R50 |
| Baseline | - | - | 54.57 | 75.02 | 64.80 | 55.36 | 85.16 | 91.89 | 98.15 |
| WRF4CIR$_{RWP}$ | - | ✓ | 56.02 | 76.34 | 66.18 | 56.78 | 85.69 | 91.86 | 98.32 |
| **WRF4CIR** | ✓ | ✓ | **57.35** | **77.33** | **67.34** | **57.85** | **86.82** | **92.70** | **98.51** |

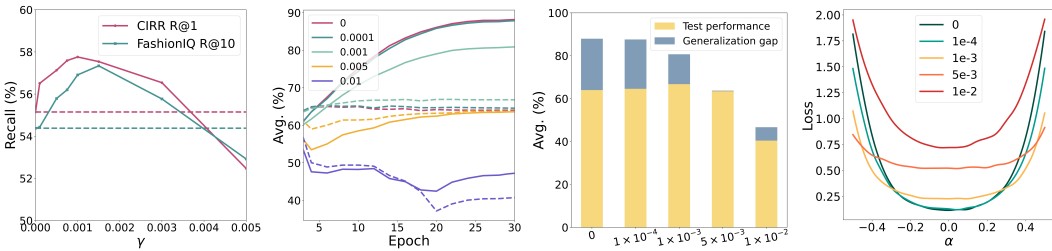

(a) Strength of weight perturbation    (b) Learning curve    (c) Generalization gap    (d) Weight loss landscape

Figure 4: **The ablation studies of WRF4CIR.** (a) Effect of perturbation strength $\gamma$ on test performance; (b), (c), and (d) Visualization of the learning curve, generalization gap, and weight loss landscape under different perturbation strength $\gamma$ on FashionIQ.

when $\gamma$ is around 0.001. Moreover, the impact of $\gamma$ on model performance typically follows a consistent rise-then-fall trend, with a noticeable improvement observed within the same range. Detailed results and analyses are provided in Appendix A.4.

Then we investigate the effect of *perturbation direction* (PD) by comparing WRF4CIR with a variant that applies random weight perturbations sampled from $\mathcal{N}(0, 1)$, denoted as WRF4CIR$_{RWP}$. The results are shown in Table 3. It can be seen that both the strength and direction of weight perturbations influence model performance. Compared to the basic framework introduced in Section 4, WRF4CIR$_{RWP}$ achieves improvements of 1.38% and 1.42% on FashionIQ Rmean and CIRR R@1, respectively. When the perturbation direction is further constrained to oppose the model gradient, performance improves even more. For example, WRF4CIR outperforms WRF4CIR$_{RWP}$ by 1.16% on FashionIQ Rmean and 1.07% on CIRR R@1.

**Visualization.** To further demonstrate the effectiveness of WRF4CIR, we visualize the *learning curve*, *weight loss landscape*, and *generalization gap* on the FashionIQ dataset. As shown in Figure 4(b), without any weight regularization, the model reaches its peak performance on the test set around the 5th epoch, after which the performance stops improving. When the perturbation strength gradually increases, the peak test performance occurs at later epochs, and the test curve more closely aligns with the training curve. Figure 4(c) illustrates the generalization gap at epoch 30. We observe that as $\gamma$ increases, the generalization gap consistently decreases, suggesting that our method leads to improved generalization.

Furthermore, we visualize the weight loss landscape of the model at epoch 30 (implementation details are provided in Appendix A.3), which characterizes the model's robustness to weight perturbations. Generally, a flatter weight loss landscape indicates better robustness to weight perturbations. As shown in Figure 4(d), when $\gamma$ increases from 0 to 0.005, the weight loss landscape becomes progressively flatter. Notably, the best-performing model at $\gamma = 0.001$ exhibits both a flat loss landscape and a low training loss. This suggests that our method effectively flattens the loss landscape without significantly impairing the training process. These results collectively demonstrate that WRF4CIR enhances model generalization by explicitly flattening the weight loss landscape, ultimately leading to improved test performance.

**Effectiveness on different CIR mechanisms.** To evaluate the effectiveness of weight-regularized fine-tuning across different CIR mechanisms, we conduct experiments using three fusion strategies under two pre-trained models (CLIP and BLIP-2). As shown in Figure 5(a), our method results in consistent improvement across different CIR mechanisms on the FashionIQ and CIRR datasets. Our method's effectiveness is also validated on two baseline approaches, including CLIP4CIR (Baldrati et al., 2023) and SPRC (Bai et al., 2024). The experimental results and implementation details

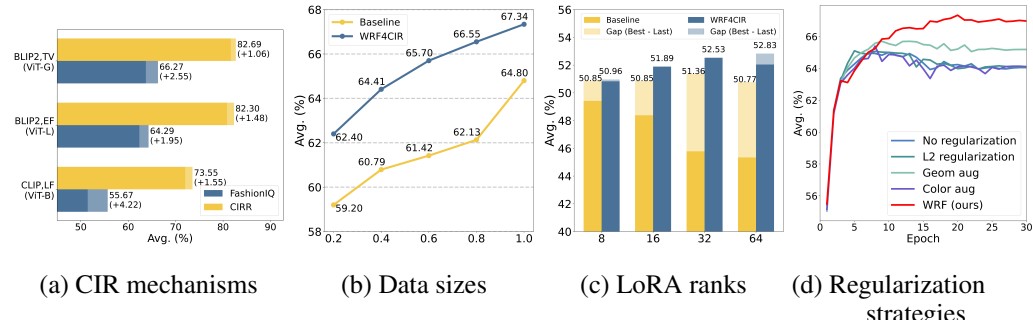

|(a) CIR mechanisms | (b) Data sizes | (c) LoRA ranks | (d) Regularization strategies |

Figure 5: Evaluation of WRF4CIR under different CIR mechanisms, training data sizes, Lora ranks and regularization strategies.

are provided in Appendix A.1. This further highlights that overfitting is a general and widespread phenomenon in VLP-based CIR and validates the effectiveness of the proposed method across different pre-trained models and fusion strategies.

**Analysis on data size.** Then, we evaluate WRF4CIR across different data sizes. Specifically, we divide the FashionIQ training set into different proportions: {0.2, 0.4, 0.6, 0.8, 1.0}, while keeping the validation set unchanged, and compare average recall on the FashionIQ val set. As shown in Figure 5(b), the proposed weight-regularized fine-tuning network is effective across varying data sizes, yielding performance gains of {+3.2, +3.62, +4.28, +4.42, +2.54}, respectively. When using only 40% of the original dataset, WRF4CIR achieves test performance close to full-data standard fine-tuning. These experimental results demonstrate that our approach can effectively fine-tune pre-trained models with limited CIR training data.

**Effectiveness on parameter-efficient fine-tuning.** To study the effectiveness of WRF4CIR under different fine-tuning strategies, we also consider LoRA-based fine-tuning (Hu et al., 2022) in addition to the standard full-parameter setting commonly used in CIR. Specifically, we adopt CLIP as the backbone and conduct experiments across different ranks on the FashionIQ and CIRR datasets. The results on FashionIQ are shown in Figure 5(c), where "best" refers to the performance of the checkpoint selected via early stopping based on validation retrieval performance during training, and "last" denotes the performance of the checkpoint from the final training epoch. As shown in the figure, WRF4CIR not only improves the best retrieval accuracy but also significantly reduces the gap between the last and best results. Additional results on different datasets are provided in Appendix A.5.

**Comparison with different regularization strategies.** We conducted ablation studies to compare WRF (ours) with commonly used regularization strategies, including L2 regularization and data augmentation. Among them, the data augmentation comprises Geom-Aug (e.g., Cutout (DeVries and Taylor, 2017)) and Color-Aug (e.g., ColorJitter). As shown in Figure 5(d), WRF consistently outperforms both L2 regularization and data augmentation methods. These ablation studies confirm that WRF provides better regularization and generalization when fine-tuning pre-trained models. Further experimental results and analyses are provided in Appendix A.7.

**Computational cost and efficiency.** Overfitting is a prevalent phenomenon in VLP-based CIR, which severely degrades model performance. To address this issue, we introduce a weight-regularized fine-tuning strategy that alleviates overfitting but requires additional computation for generating adversarial perturbations, increasing the training time from 2.85 h to 4.78 h over 30 epochs. Notably, this overhead can be largely mitigated by adopting random weight perturbation, which reduces the training time to 2.95 h. With negligible additional computational cost, our method with random weight perturbation achieves significant performance gains, as shown in Table 3. More importantly, our method is applied only during fine-tuning, thereby introducing no additional inference overhead.

## 6 CONCLUSION

In this work, we find that overfitting is a prevalent issue in VLP-based Composed Image Retrieval (CIR), which has become a major bottleneck for further performance improvement. To address this problem, we introduce a Weight-Regularized Fine-tuning network for CIR, termed WRF4CIR. It

regularizes the fine-tuning process by introducing adversarial perturbations to the model weights. Extensive experiments demonstrate that our method significantly narrows the generalization gap and achieves substantial improvements on benchmark datasets, validating the effectiveness of the proposed approach.

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

# A APPENDIX

The following items are included in our supplementary material:

- Details about Mechanisms of CIR Methods in Appendix A.1.
- More Results on the Impact of Data Size in Appendix A.2.
- Visualization of Weight Loss Landscape in Appendix A.3.
- More Analysis on Perturbation Strength $\gamma$ in Appendix A.4.
- Weight-Regularized Parameter-Efficient Fine-Tuning in Appendix A.5.
- Broader Impact in Appendix A.6.
- Comparison with Different Regularization Strategies in Appendix A.7.
- Computational Cost and Efficiency in Appendix A.8.
- Visualization of Retrieval Results in Appendix A.9.

## A.1 DETAILS ABOUT MECHANISMS OF CIR METHODS

CIR mechanisms involve the choice of pre-trained models and fusion strategies, which have been explored in prior work (Baldrati et al., 2023; Bai et al., 2024; Liu et al., 2024; Jiang et al., 2024). In general, powerful pre-trained models combined with appropriate fusion strategies contribute to better model performance. In Section 3, we investigate the overfitting problem in VLP-based CIR. To assess the generality of this issue, we visualize the learning curves of different CIR mechanisms based on the following three fusion strategies:

**Late-fusion** integrates image and text features after the feature encoders. A simple implementation utilizes CLIP as a feature encoder, deriving a multimodal representation by directly summing the extracted features, e.g., CLIP4CIR(sum) (Baldrati et al., 2023).

**Early-fusion** facilitates the interaction between image and text representations during feature extraction, capturing cross-modal correlations more effectively. Our WRF4CIR adopts BLIP-2 as its backbone and leverages its image-text matching encoder to implement early fusion, effectively keeping a balance between late fusion and input-level fusion.

**Sentence-level text inversion** enables input-level fusion by mapping image representation into the word embedding space of VLP models. It utilizes the alignment ability of VLPs and improves performance of CIR (Bai et al., 2024). The pipeline involves projecting an image into a set of pseudo-word tokens, which are subsequently concatenated with tokenized text inputs before being processed by the text encoder.

The overfitting phenomenon and generalization gap are illustrated in Figure 1, while the effectiveness of WRF4CIR across various CIR mechanisms is reported in Figure 5 and Table 4. Additionally, the effectiveness of our method is further validated on different baselines, including CLIP4CIR (Baldrati et al., 2023) and SPRC (Bai et al., 2024), with the corresponding results on FashionIQ dataset reported in Table 4.

| Method | Backbone | Dress | | Shirt | | TopTee | | Average | | |
|---|---|---|---|---|---|---|---|---|---|---|
| | | R10 | R50 | R10 | R50 | R10 | R50 | R10 | R50 | Rmean |
| CLIP,LF | ViT-B | 35.29 | 59.99 | 40.13 | 61.38 | 43.80 | 68.12 | 39.74 | 63.16 | 51.45 |
| CLIP,LF+Ours | | **41.2** | **65.15** | **43.28** | **65.06** | **47.68** | **71.34** | **44.05** | **67.18** | **55.61** |
| CLIP4CIR(combiner) | ViT-B | 37.72 | 64.35 | 43.81 | 65.40 | 47.06 | 70.57 | 42.87 | 66.77 | 54.82 |
| CLIP4CIR(combiner)+Ours | | **43.77** | **66.98** | **46.46** | **67.22** | **51.60** | **73.58** | **47.28** | **69.26** | **58.27** |
| BLIP-2,EF | ViT-L | 46.01 | 70.4 | 52.4 | 72.72 | 55.89 | 76.39 | 51.43 | 73.16 | 62.34 |
| BLIP-2,EF+Ours | | **49.67** | **72.78** | **54.07** | **74.14** | **56.85** | **77.81** | **53.68** | **74.91** | **64.29** |
| BLIP-2,TV | ViT-G | 46.27 | 71.06 | 53.62 | 73.3 | 59.11 | 78.95 | 53 | 74.44 | 63.72 |
| BLIP-2,TV+Ours | | **51.66** | **74.31** | **56.62** | **75.66** | **59.61** | **79.75** | **55.96** | **76.57** | **66.27** |
| SPRC | ViT-G | 49.18 | 72.43 | 55.64 | 73.89 | 59.35 | 78.58 | 54.92 | 74.97 | 64.85 |
| SPRC+Ours | | **52.35** | **75.27** | **57.45** | **76.30** | **60.22** | **79.85** | **56.67** | **77.12** | **66.90** |

Table 4: **Results on different CIR mechanisms.** The best results are shown in bold fonts.

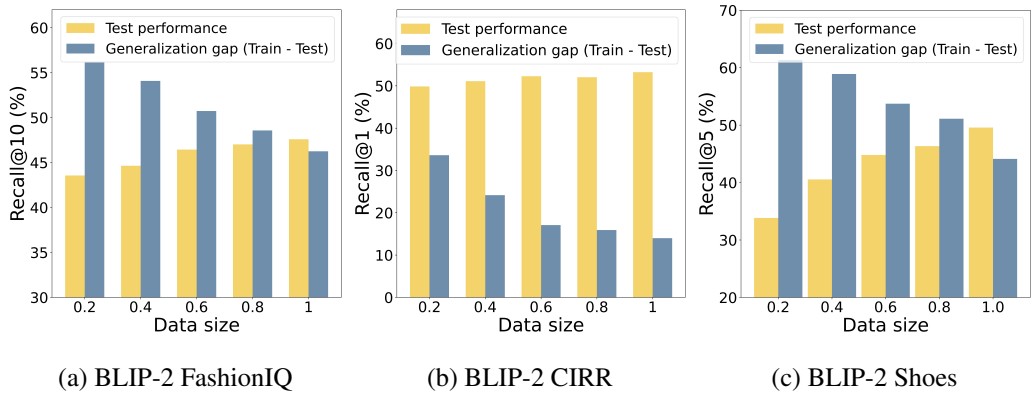

Figure 6: Generalization gap and test performance of BLIP-2 across different training data sizes.

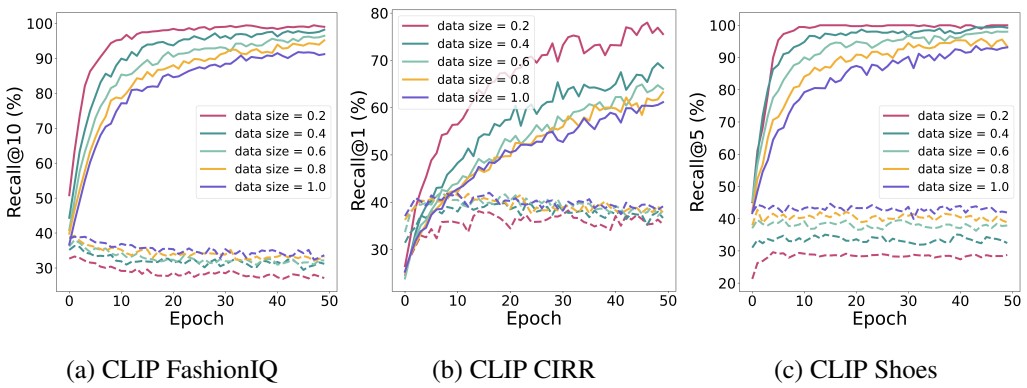

Figure 7: Learning curve of CLIP across different training data sizes.

## A.2 MORE RESULTS ON THE IMPACT OF DATA SIZE

Here, we present the impact of data size on test performance and generalization gap across different datasets and pre-trained models, as a supplementary to Figure 2. Considering that metrics such as R@10 or R@50 tend to be saturated on certain datasets—resulting in near-perfect performance on the training set regardless of data size—we report R@1 on the CIRR, R@5 on the Shoes datasets and R@10 on the FashionIQ dataset. Figure 6 show the results of BLIP-2 on the FashionIQ, CIRR and Shoes datasets. Figure 7 presents the CLIP-based experimental results on FashionIQ, CIRR, and Shoes datasets. The results show that, given the same number of training epochs, models trained with fewer data consistently achieve higher training accuracy but lower test accuracy. This leads to an increasingly large generalization gap, which is a clear indicator of overfitting.

## A.3 VISUALIZATION OF WEIGHT LOSS LANDSCAPE

In this section, we present the pseudo-code for visualizing the weight loss landscape. As shown in Algorithm 2, we first randomly sample a perturbation direction from a Gaussian distribution $\mathcal{N}(0, 1)$ and initialize the perturbation size based on the parameter scales of different layers (Line 2-5). We then scale the perturbation size using a series of scaling factors $\alpha \in \{\alpha_{min}, \dots, \alpha_{max}\}$ to evaluate the model's robustness to weight perturbations (Line 6-8). Next, we compute the contrastive loss on all normal samples using the perturbed model $f_{\theta+\alpha d}$ (Line 7). Finally, we visualize the weight loss landscape (Line 9).

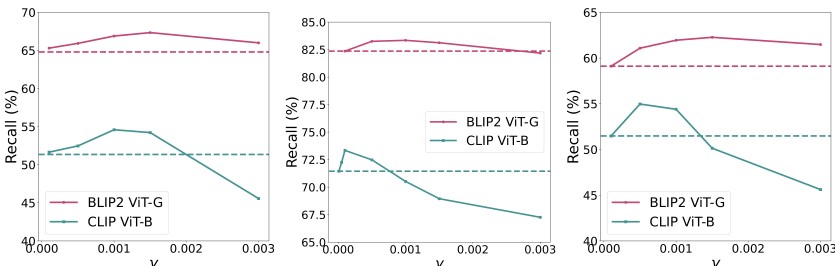

Figure 8: Effect of perturbation strength $\gamma$ across different datasets and pre-trained models. *Left*: FashionIQ; *Middle*: CIRR; *Right*: Shoes.

---

**Algorithm 2** Visualization of Weight Loss Landscape

1: **Input:** Training set $D_t = \{(I_r^i, T^i, I_t^i)\}_{i=1}^n$; the scalar parameter $\alpha \in [\alpha_{min}, \alpha_{max}]$; CIR model $f_\theta$ with L-layer.
2: Sample a direction $d \sim \mathcal{N}(0,1)$
3: **for** $l = 1, \ldots, L$ **do**
4: $\quad d_l \leftarrow \frac{d_l}{\|d_l\|} \|\theta_l\|$
5: **end for**
6: **for** $\alpha = \alpha_{min}, \ldots, \alpha_{max}$ **do**
7: $\quad \rho(\theta + \alpha d) \leftarrow \frac{1}{n} \sum_{i=1}^n \ell_{q2t}(f_{\theta+\alpha d}(T^i, I_r^i), f_{\theta+\alpha d}(I_t^i))$
8: **end for**
9: Plot $(\alpha, \rho(\theta + \alpha d)), \forall \alpha \in [\alpha_{min}, \alpha_{max}]$

---

### A.4 MORE ANALYSIS ON PERTURBATION STRENGTH $\gamma$

To ensure consistent performance gains, we additionally present effective heuristics for choosing $\gamma$. As shown in Figures 4 and 8, model performance consistently follows a rise-then-fall trend as $\gamma$ increases within the range [0, 0.003], across different datasets and models. Therefore, in practice, a simple grid search (e.g., $\gamma \in \{0.1, 0.01, 0.001, 0.0001\}$ ) is usually sufficient to identify a coarse range for the optimal $\gamma$. It is also worth noting that $\gamma$ represents the ratio between the norm of the weight perturbations and that of the model parameters. Thus, a small $\gamma$ does not necessarily imply a negligible adversarial perturbation. In our method, it is easy to identify a suitable $\gamma$. As shown in Figures 4 and 8, the optimal $\gamma$ values are highly consistent across various models and datasets. This consistency makes it easy to select a single $\gamma$ value that performs well in different experimental settings. For example, when $\gamma = 0.001$, the model achieves strong performance across most settings. These observations provide useful practical guidance for hyperparameter selection when adapting to new datasets or models.

### A.5 WEIGHT-REGULARIZED PARAMETER-EFFICIENT FINE-TUNING

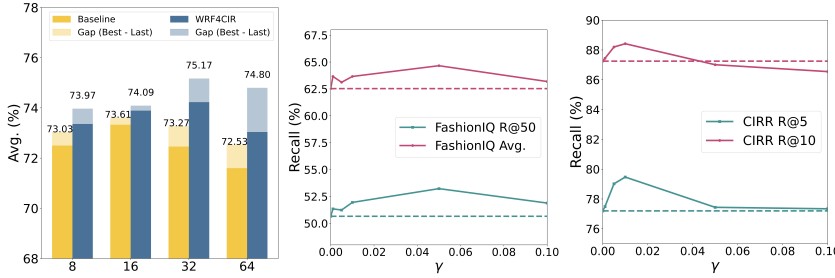

Figure 9: WRF4CIR results on CLIP for varying LoRA ranks and $\gamma$. *Left*: Different ranks on the CIRR dataset; *Middle*: Different $\gamma$ on the FashionIQ dataset (rank=8); *Right*: Different $\gamma$ on the CIRR dataset (rank=32).

Table 5: Comparison of different regularization strategies on FashionIQ and CIRR datasets.

| Method | FashionIQ | | | CIRR | | |
|---|---|---|---|---|---|---|
| | R10 | R50 | Rmean | R1 | R5 | R10 |
| No regularization | 54.42 | 75.12 | 64.67 | 55.18 | 85.41 | 91.67 |
| L2 regularization | 54.57 | 75.02 | 64.80 | 55.36 | 85.16 | 91.89 |
| Geom-Aug | 55.72 | 75.36 | 65.54 | 55.17 | 85.12 | 91.65 |
| Color-Aug | 54.63 | 75.06 | 64.84 | 54.12 | 84.93 | 91.62 |
| **WRF (ours)** | **57.35** | **77.33** | **67.34** | **57.85** | **86.82** | **92.70** |

## A.6 BROADER IMPACT

Overfitting is a common issue in VLP-based CIR and has become a bottleneck limiting model performance. In this work, we introduce WRF4CIR to improve model generalization, which may contribute to building more robust retrieval systems in real-world scenarios. However, WRF4CIR introduces additional computational overhead, which may have environmental implications, such as increased carbon emissions. At the same time, we would like to caution against over-optimism regarding the generalization capabilities of retrieval systems. The reported generalization performance is based on specific training and test datasets, while real-world retrieval systems face more diverse and unpredictable data sources.

## A.7 COMPARISON WITH DIFFERENT REGULARIZATION STRATEGIES

CIR aims to retrieve a target image given a reference image and a description text, where the text specifies the transformation from the reference to the target. For example, in the FashionIQ dataset, when the reference and target images depict clothing items of the same style but different colors, certain data augmentations such as ColorJitter can distort the color information and thus disrupt the semantic correspondence. Similarly, Geom-Aug (e.g., Cutout) may alter spatial details in a way that causes inaccurate matches.

We conducted ablation studies to evaluate the isolated effects of different regularization strategies. Among them, the data augmentation comprises Geom-Aug (e.g., Cutout) and Color-Aug (e.g., ColorJitter). As shown in Table 5, WRF consistently outperforms both L2 regularization and data augmentation methods on the FashionIQ and CIRR benchmarks. These ablation studies confirm that WRF provides better regularization and generalization in the CIR task. Figure 10 shows additional comparisons of regularization methods and their results.

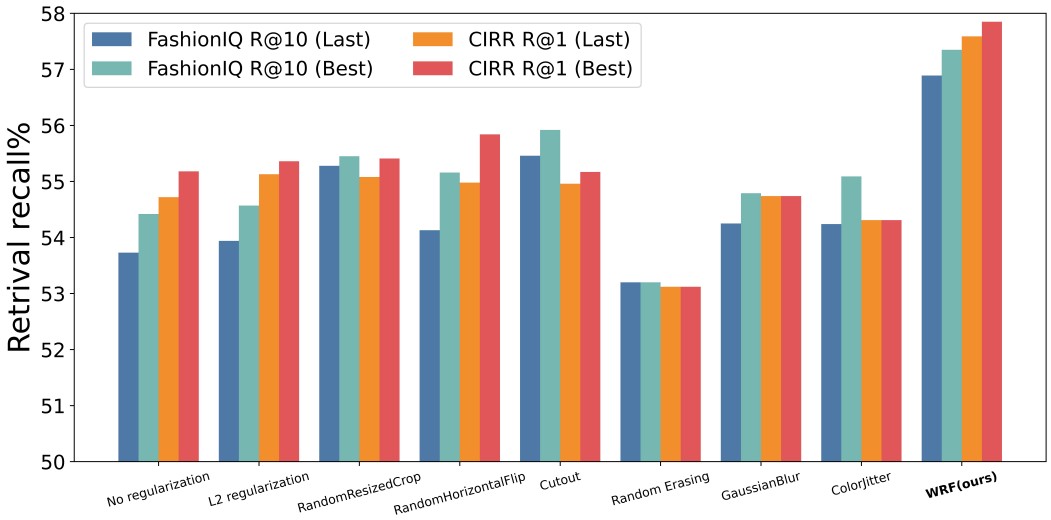

Figure 10: Additional comparisons of regularization strategies on the FashionIQ and CIRR datasets.

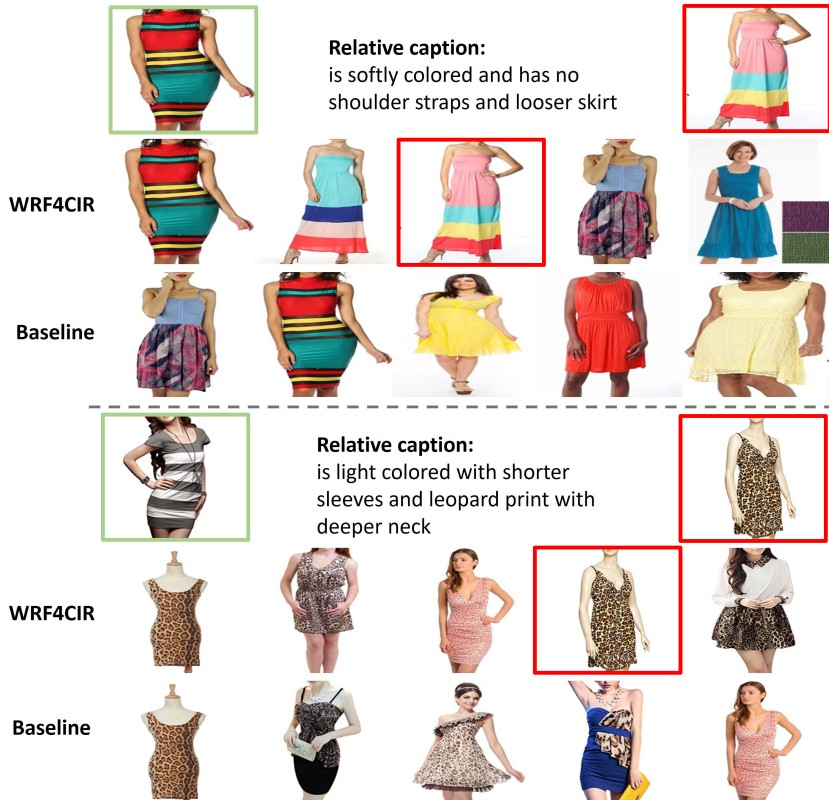

Figure 11: Retrieval results of WRF4CIR with respect to the Recall@5 metric on the FashionIQ dataset, where the reference image and ground-truth retrieval images are highlighted with green and red outlines. The retrieval results in the second row are ranked from left to right as rank 1 to rank 5, respectively.

### A.8    COMPUTATIONAL COST AND EFFICIENCY

Our method incurs additional training overhead due to the use of a second backpropagation, but it introduces no extra inference cost. Considering the performance gains, the overall computational burden is acceptable. This overhead stems from the use of double backpropagation during training. On FashionIQ with an A100 GPU, training without WRF requires **2.85** hours for 30 epochs, plus **1.29** hours for validation and early stopping due to overfitting, totaling **4.14** hours. In contrast, WRF4CIR stabilizes convergence and reduces validation time. For BLIP-2, the second backpropagation bypasses the frozen image encoder, and the lightweight Q-Former further limits overhead, resulting in **4.78** hours for 30 epochs.

To further reduce training overhead, we introduce a variant of random weight perturbation that avoids double backpropagation. Although this variant slightly reduces the accuracy of the retrieval, it decreases the total training time to **2.95** hours for 30 epochs.

### A.9    VISUALIZATION OF RETRIEVAL RESULTS

Here, we present several visualized retrieval results on the FashionIQ and CIRR datasets. Each sample consists of a reference image, a textual description, and the top-5 retrieved images. As shown in Figure 12, on the open-domain CIRR dataset, WRF4CIR accurately retrieves the target image compared to its variant without weight-regularized fine-tuning, demonstrating its effectiveness in integrating information from both the reference image and the textual description. Figure 11 shows the results on the FashionIQ dataset. Our method demonstrates superior R@5 recall and a strong ability to accurately capture complex semantic concepts in the textual queries, such as 'leopard print with deeper neck'.

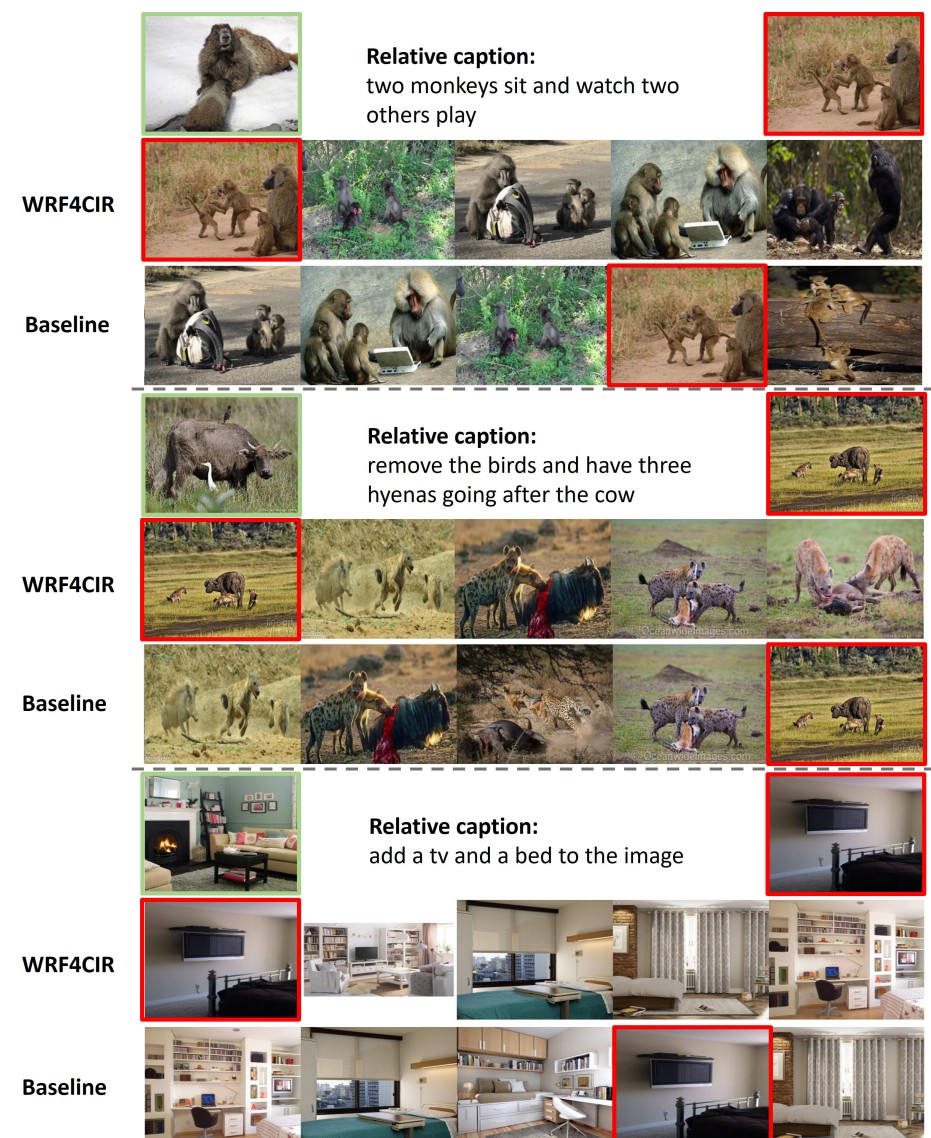

Figure 12: Retrieval results of WRF4CIR with respect to the Recall@1 metric on CIRR dataset, where the reference image and ground-truth retrieval images are highlighted with green and red outline. The retrieval results in the second row are ranked from left to right as rank 1 to rank 5, respectively.

