# OpenReview forum: "WRF4CIR: Weight-Regularized Fine-Tuning Network for Composed Image Retrieval"
_ICLR.cc/2026/Conference — Submitted to ICLR 2026_

### Official Review · Reviewer_5Ffw · 2025-10-19

**Soundness:** 3
**Presentation:** 3
**Contribution:** 3
**Rating:** 4
**Confidence:** 5

**Summary:**

This paper proposes a novel Weight-Regularized Fine-tuning network for CIR, termed WRF4CIR.  By introducing adversarial perturbations to the model weights, which act inversely to gradient descent, WRF4CIR increases the difficulty of learning the model on the training data, effectively mitigating overfitting. Extensive experiments on benchmark datasets demonstrate that WRF4CIR significantly narrows the generalization gap and achieves significant improvement over existing methods.

**Strengths:**

1. The idea of ​​providing perturbations to the parameters in the direction opposite to the gradient is interesting.
2. There are sufficient comparative tests and experimental analysis

**Weaknesses:**

1. Regarding the overfitting issue, as the training data decreases, it may not be overfitting, but insufficient training. I think this part of the analysis is not sufficient.
2. I think the starting point of this article is good, but the analysis and method design need to be improved.
3. How much improvement does Q-Former bring to the method? It seems that Q-Former plays an important role in the method, but the ablation experiment does not provide corresponding results.

**Questions:**

see weakness

---

> ### Author Response · Authors · 2025-11-20
> **PART 1**
>
> We sincerely appreciate your time and effort in reviewing our manuscript. Below, we address your main concerns of this work.
>
> > **Q1**: Regarding the overfitting issue, as the training data decreases, it may not be overfitting, but insufficient training. I think this part of the analysis is not sufficient.
>
> **A1**: Thank you for raising this important concern. To distinguish overfitting from insufficient training, we conducted a detailed analysis by visualizing learning curves under different training data sizes and examining the change of both training and test performance. Below, we provide representative results at epochs 10, 25, and 50 to illustrate the trend, and the full learning curves of three datasets are provided in Figure 7 of the updated manuscript.
>
> **Table 1**: FashionIQ Recall@10 performance of different training data sizes. (Val set / Training set)
>
> | **Epoch**    | **Data size 0.2** | **Data size 0.4** | **Data size 0.6** | **Data size 0.8** | **Data size 1** |
> | ------------ | ----------------- | ----------------- | ----------------- | ----------------- | --------------- |
> | **epoch=10** | 29.15 / 95.35     | 31.84 / 89.91     | 34.37 / 85.35     | 35.19 / 78.53     | 36.89 / 77.14   |
> | **epoch=25** | 28.18 / 98.17     | 31.24 / 95.81     | 31.43 / 92.45     | 34.31 / 89.72     | 34.45 / 87.45   |
> | **epoch=50** | 27.17 / 99.02     | 31.18 / 98.19     | 32.14 / 96.45     | 33.21 / 95.15     | 33.64 / 91.19   |
>
> The results show that, given the same number of training epochs, models trained with fewer data points consistently achieve higher training accuracy but lower test accuracy. This leads to an increasingly large generalization gap, which is a clear indicator of overfitting rather than insufficient training.
>
> We hope this clarification addresses your concern, and we would be happy to provide further discussion if needed.
>
> ---
>
> > **Q2**: I think the starting point of this article is good, but the analysis and method design need to be improved.
>
> **A2**: We appreciate the acknowledgement of the motivation and starting point of our work. Since the comment is high-level, we kindly welcome more specific suggestions regarding which aspects of the analysis or method design the reviewer finds insufficient. We would be happy to further refine any part of the work based on more detailed guidance from the reviewer.

---

> > ### Author Response · Authors · 2025-11-20
> > **PART 2**
> >
> > > **Q3**: How much improvement does Q-Former bring to the method? It seems that Q-Former plays an important role in the method, but the ablation experiment does not provide corresponding results.
> >
> > **A3**: Thank you for the insightful question. Q-Former is an inherent component of the BLIP-2 architecture, which serves as the backbone in our method. Since our method is designed as a fine-tuning strategy built upon BLIP-2, we follow its standard architecture and do not modify or remove Q-Former.
> >
> > For this reason, Q-Former is not an independent design choice introduced by our approach but part of the underlying model we build on. However, we agree that understanding the backbone dependency is valuable. In our comparative experiments, we compare our method with recent works that adopt the same backbone, such as SPRC [1], CCIN [2], and TME [3], and the results are presented in Tables 1 and 2. We also evaluated WRF4CIR under different CIR mechanisms (late fusion, early fusion, and text inversion). The results, provided in Figure 5(a) and Table 4, show that our method is effective across various fusion strategies and pretrained backbones.
> >
> > To further address your concern, we validate our method on existing approaches (e.g., CLIP4CIR+Ours, SPRC+Ours). We observe that our method consistently brings performance improvements. These results support that the effectiveness of WRF4CIR does not rely on the presence of Q-Former.
> >
> > |            Method             |     Backbone     | Dress R@10 | Dress R@50 | Shirt R@10 | Shirt R@50 | TopTee R@10 | TopTee R@50 | Avg R@10  | Avg R@50  |   Rmean   |
> > | :---------------------------: | :--------------: | :--------: | :--------: | :--------: | :--------: | :---------: | :---------: | :-------: | :-------: | :-------: |
> > |         CLIP4CIR(sum)         |    CLIP ViT-B    |   35.29    |   59.99    |   40.13    |   61.38    |    43.80    |    68.12    |   39.74   |   63.16   |   51.45   |
> > |   **CLIP4CIR(sum) + Ours**    |  **CLIP ViT-B**  | **41.20**  | **65.15**  | **43.28**  | **65.06**  |  **47.68**  |  **71.34**  | **44.05** | **67.18** | **55.61** |
> > |      CLIP4CIR(combiner)       |    CLIP ViT-B    |   37.72    |   64.35    |   43.81    |   65.40    |    47.06    |    70.57    |   42.87   |   66.77   |   54.82   |
> > | **CLIP4CIR(combiner) + Ours** |  **CLIP ViT-B**  | **43.77**  | **66.98**  | **46.46**  | **67.22**  |  **51.60**  |  **73.58**  | **47.28** | **69.26** | **58.27** |
> > |             SPRC              |   BLIP-2 ViT-G   |   49.18    |   72.43    |   55.64    |   73.89    |    59.35    |    78.58    |   54.92   |   74.97   |   64.85   |
> > |        **SPRC + Ours**        | **BLIP-2 ViT-G** | **52.35**  | **75.27**  | **57.45**  | **76.30**  |  **60.22**  |  **79.85**  | **56.67** | **77.12** | **66.90** |
> >
> > These additional results and the corresponding descriptions have been included in Table 4 and Appendix A1 of the updated manuscript. We hope this clarification addresses your concern, and we would be happy to provide further discussion if needed.
> >
> > **Reference:**
> >
> > [1] Sentence-level Prompts Benefit Composed Image Retrieval. ICLR 2024.
> > [2] CCIN: Compositional Conflict Identification and Neutralization for Composed Image Retrieval. CVPR 2025.
> > [3] Learning with Noisy Triplet Correspondence for Composed Image Retrieval. CVPR 2025.

---

> > ### Comment · Reviewer_5Ffw · 2025-11-24
> >
> > 1. Regarding the author's motivation, " first to reveal that overfitting is a prevalent phenomenon in VLP-based CIR," there were no specific improvements made to the VLP model; it appears that weight regularization was simply applied to the CIR task.
> > 2. Following on to question 1, the authors' method does not offer specific improvements for VLP or CIR tasks. It appears to be applicable to other similar tasks that require preventing overfitting, such as image retrieval and image-text retrieval.
> > 3. I would love to see the author's work compared to other REGULARIZATION methods.
> > In conclusion, I believe the authors' work still needs improvement, such as making improvements to address the characteristics of VLPs or conducting experiments on non-VLP models to demonstrate generalization.

---

> ### Author Response · Authors · 2025-11-24
>
> Thank you very much for your thoughtful feedback. Below, we provide clarifications addressing each of your concerns.
>
> **1. On the motivation and whether our work offers specific improvements for VLP-based CIR**
>
> Revealing the overfitting phenomenon in VLP-based CIR is itself an important contribution. Our work demonstrates that overfitting has a substantial impact on CIR performance, yet this issue has been largely overlooked. Specifically, we identify a significant generalization gap in VLP fine-tuning for CIR, which lays the foundation for addressing this problem. Building on this analysis, we introduce a weight-regularized fine-tuning strategy that effectively mitigates overfitting and achieves notable improvement over recent methods. Our work thus provides an efficient approach to VLP-based CIR under limited data.
>
> Beyond the observed performance gains, our analysis of overfitting carries broader implications for CIR research. Due to the severity of overfitting, many potentially effective CIR methods may fail to exhibit their true capability simply because they are not sufficiently trained. By identifying this limitation and providing an effective remedy, our work enables more reliable and stable evaluations within the community. This contributes to establishing a more faithful training paradigm for future research on VLP-based CIR. We believe that both the analysis of overfitting and the proposed mitigation strategy offer meaningful insights to the CIR community.
>
> **2. On whether our method is a simple application of weight regularization and comparison with other regularization approaches**
>
> Our method is inspired by AWP [1]. We adapt weight-perturbation–based regularization specifically to the CIR task, taking into account several key differences, including:
>
> - **Task:** CIR retrieval vs. image classification
> - **Loss function:** contrastive loss vs. cross-entropy
> - **Training scenario:** fine-tuning large VLP models vs. training CNNs from scratch
> - **Data format:** multimodal triplets (reference image, modification text, target image) vs. single-modality images
>
> These differences require nontrivial adjustments, making the method specialized rather than a direct transfer.
>
> Regarding comparisons with other regularization methods, Section 5.3 already includes experiments comparing our approach with commonly used regularization techniques. The results in Figure 5(d) show that WRF consistently outperforms both L2 regularization and data augmentation, demonstrating that WRF provides better regularization and generalization in the CIR task.
>
> **3. On the applicability of our method to other tasks such as image retrieval and image–text retrieval**
>
> We agree that the potential to extend our method to other tasks is an interesting direction, but we do not claim that the method will necessarily be effective. For example, although data augmentation is widely used to combat overfitting in many tasks, we find that standard augmentation provides little benefit for VLP-based CIR. Exploring extensions to other domains is promising but remains an open research question, and we appreciate the reviewer for highlighting this direction.
>
> We hope this clarification addresses your concern, and we would be happy to provide further discussion if needed.
>
> **Reference:**
>
> [1] Adversarial weight perturbation helps robust generalization. NIPS 2020.

---

> > ### Comment · Reviewer_5Ffw · 2025-11-24
> >
> > 1. The difference between Eq.4 and AWP is insufficient to support the work of the entire paper.
> > 2. The explanation regarding weight regularization is weak and inadequate; it only reflects differences in the tasks, not the innovation of the proposed method. As you said, you only replaced the original cross-entropy with contrastive loss.
> > 3. Returning to the contrastive loss, InfoNCE is also a combination of contrastive loss and cross-entropy loss.
> > 4. The regularization methods compared are all quite simple; I think we need to compare them with more published regularization methods.
> >
> > The experimental results are indeed sufficient, but if the authors' innovation is limited to improvements on AWP, I think this is more suitable as a technical report than an academic paper for publication.

---

> > > ### Author Response · Authors · 2025-11-25
> > >
> > > Thank you for the thoughtful follow-up and for acknowledging the solidity of our experiments. We address your remaining concerns as follows.
> > >
> > > **1. On the differences between Eq. 4 and AWP**
> > >
> > > We would like to clarify that we never claimed that our method “only replaces cross-entropy with contrastive loss.” In Eq. 4, beyond the loss function $ℓ_{q2t}$, several key components fundamentally differ from AWP:
> > >
> > > - the model $f_\theta$ (a large-scale VLP model rather than a CNN),
> > > - the optimization setting (fine-tuning vs. training from scratch),
> > > - the multimodal triplet input ($I_r$, $T$, $I_t$) rather than single-modality data.
> > >
> > > These differences coexist and collectively define a substantially different problem setting.
> > >
> > > **2. On the novelty of our work**
> > >
> > > Our work demonstrates that weight-perturbation-based regularization is effective for mitigating overfitting in VLP-based CIR—a direction that, to the best of our knowledge, has not been explored within the CIR community. Moreover, we are the first to reveal that overfitting is a prevalent phenomenon in VLP-based CIR. Therefore, we respectfully disagree that “the innovation is limited to improvements on AWP”.
> > >
> > > **3. On comparisons with additional regularization methods**
> > >
> > > We have compared our method against several published data augmentation approaches on the CIRR dataset, including TrivialAugment [1], AugMix [2], Cut Thumbnail [3], and Self Augmentation [4]. As shown in the results, our method significantly outperforms all these techniques.
> > >
> > > | Method                                 | R@1               | R@5               | R@10              | R@50              | AVG               |
> > > | -------------------------------------- | ----------------- | ----------------- | ----------------- | ----------------- | ----------------- |
> > > | Baseline                               | 55.36             | 85.16             | 91.89             | 98.15             | 82.14             |
> > > | TrivialAugment (num_magnitude_bins=5)  | 55.53 (+0.17)     | 85.38 (+0.22)     | 91.77 (-0.12)     | 98.13 (-0.02)     | 82.20 (+0.06)     |
> > > | TrivialAugment (num_magnitude_bins=10) | 55.48 (+0.12)     | 85.14 (-0.02)     | 91.89 (+0.00)     | 98.06 (-0.09)     | 82.14 (+0.00)     |
> > > | TrivialAugment (num_magnitude_bins=20) | 55.34 (-0.02)     | 84.97 (-0.19)     | 91.55 (-0.34)     | 98.15 (+0.00)     | 82.01 (-0.13)     |
> > > | AugMix(severity=3)                     | 55.63 (+0.27)     | 85.24 (+0.08)     | 92.03 (+0.14)     | 98.06 (-0.09)     | 82.24 (+0.10)     |
> > > | AugMix(severity=6)                     | 55.75 (+0.39)     | 85.45 (+0.29)     | 91.77 (-0.12)     | 98.15 (+0.00)     | 82.28 (+0.14)     |
> > > | AugMix(severity=9)                     | 55.94 (+0.58)     | 84.97 (-0.19)     | 91.93 (+0.04)     | 98.13 (-0.02)     | 82.24 (+0.10)     |
> > > | Cut Thumbnail                          | 55.27 (-0.09)     | 85.05 (-0.11)     | 91.86 (-0.03)     | 97.96 (-0.19)     | 82.03 (-0.11)     |
> > > | Self Augmentation                      | 55.51 (+0.15)     | 85.41 (+0.25)     | 92.10 (+0.21)     | 98.01 (-0.14)     | 82.26 (+0.12)     |
> > > | **Ours**                               | **57.85 (+2.49)** | **86.82 (+1.66)** | **92.70 (+0.81)** | **98.51 (+0.36)** | **84.72 (+2.58)** |
> > >
> > > We hope the above clarifications address your concerns and would be happy to further elaborate if needed.
> > >
> > > **Reference:**
> > >
> > > [1] Trivialaugment: Tuning-free yet state-of-the-art data augmentation. ICCV 2021.
> > > [2] Augmix: A simple data processing method to improve robustness and uncertainty. ICLR 2020.
> > > [3] Cut-thumbnail: A novel data augmentation for convolutional neural network." ACMMM 2021.
> > > [4] Selfaugment: Automatic augmentation policies for self-supervised learning. CVPR 2021.

---

> > > > ### Author Response · Authors · 2025-11-27
> > > >
> > > > Thank you again for your thoughtful engagement during the discussion stage. We hope everything is going smoothly on your end. If there are any additional points you would like us to clarify or elaborate on, we would be very happy to provide further details. Please feel free to let us know at your convenience.

---

> > > > ### Comment · Reviewer_5Ffw · 2025-11-27
> > > >
> > > > 1. The authors outlined the differences between their method and AWP from the perspectives of model and initialization. However, they did not propose solutions to these differences, except for modifications to the loss function.
> > > > 2. AWP attempts to improve the robustness of generalization, which is similar to your approach to combating overfitting.
> > > > 3. Another puzzling issue is that the authors consistently point to the difference between VLP and CNN models, but no solution to this difference has been found.
> > > >
> > > > I am very grateful for the authors' patient and detailed response. After carefully reading the response, I have decided to maintain my original score.

---

> > > > > ### Author Response · Authors · 2025-11-27
> > > > >
> > > > > Thank you very much for your prompt response. We address your remaining concerns in detail below.
> > > > >
> > > > > If we understand correctly, the central question is whether the differences between our method and AWP meaningfully extend beyond the loss modification. To clarify this, we provide a more systematic comparison, including implementation-level distinctions and similarities.
> > > > >
> > > > > **Differences from AWP**
> > > > >
> > > > > - **Loss function.** As the reviewer noted, our method uses a contrastive loss, whereas AWP relies on cross-entropy.
> > > > > - **Perturbation target.** Our method perturbs the **Q-Former (Transformer) and learnable queries** within BLIP-2—a multimodal VLP architecture. In contrast, AWP perturbs **convolutional layers** in CNNs.
> > > > > - **Perturbation scope.** Our approach applies perturbation to **a selected subset of BLIP-2 parameters**, while AWP perturbs **the entire network**.
> > > > >
> > > > > These differences are nontrivial because VLPs use fundamentally different architectures, optimization dynamics, and data modalities compared to CNNs. Consequently, adapting weight-perturbation–based regularization to VLP-based CIR requires meaningful changes to the design and implementation.
> > > > >
> > > > > **Similarities with AWP**
> > > > >
> > > > > - The **perturbation strategy** and **perturbation constraint** follow the adversarial perturbation formulation introduced in AWP, and we have clearly acknowledged and cited this connection.
> > > > >
> > > > > **Additional empirical evidence**
> > > > >
> > > > > Beyond the architectural and methodological differences, we would also like to highlight an important empirical distinction:
> > > > >
> > > > > - In AWP, **replacing adversarial perturbation with random perturbation eliminates the performance gain**.
> > > > > - In our work, **random perturbation still brings a significant improvement in CIR performance**.
> > > > >
> > > > > This observation reveals a new insight: **the effectiveness of weight perturbation in VLP-based CIR is not solely tied to adversarial optimization**, suggesting different underlying mechanisms compared to AWP.
> > > > >
> > > > > Hope the clarification above could address your concerns. We sincerely appreciate the reviewer’s time and careful evaluation, and please let us know if there is more to clarify.

---

### Official Review · Reviewer_11Dd · 2025-10-26

**Soundness:** 3
**Presentation:** 3
**Contribution:** 2
**Rating:** 2
**Confidence:** 4

**Summary:**

The paper aims to address the significant overfitting problem that occurs when fine-tuning VLPs on CIR tasks, which often have limited training data. The authors propose a weight-regularized fine-tuning method called WRF4CIR, which applies adversarial perturbations to the model weights to flatten the loss landscape and, consequently, improve generalization .

**Strengths:**

The motivation is clear and the experimental results show some improvement on the FashionIQ and CIRR datasets.

**Weaknesses:**

1. The primary weakness of this paper is the positioning of its core contribution. The authors claim to propose a "novel" Weight-Regularized Fine-tuning network, with the central idea of finding "flatter minima" in the loss landscape to improve generalization.This concept is not new; it is extensively studied in the deep learning community. More importantly, the proposed optimization objective (Eq 1 7, Eq 3), $\min_{\theta} \max_{\delta} \mathcal{L}(\theta+\delta)$, is functionally identical to the core mechanism of Sharpness-Aware Minimization (SAM) (Foret et al., 2021) and its variants (like ASAM). SAM is a highly prominent and widely used regularization technique that explicitly seeks flat loss regions by finding adversarial perturbations (via gradient ascent) in the weight space. Critically, the paper fails to cite or even mention SAM anywhere. WRF4CIR appears to be a direct application of SAM (or a very similar adversarial weight perturbation technique) to the CIR task.
2. The paper's optimization objective (Eq 3) is a $\min \max$ problem. This requires an inner loop of gradient ascent to find the perturbation $\delta$ that maximizes the loss. The authors correctly state this, noting the perturbation is "generated in the opposite direction of gradient descent". The direction of gradient descent is $-\nabla \mathcal{L}$, so its opposite (gradient ascent) should be $+\nabla \mathcal{L}$. However, in Algorithm 1, Line 5, the formula given for the perturbed model is:$f_{\theta+\delta}(t)\leftarrow f_{\theta}(t)-\gamma\frac{L(f)}{||L(f)||}||f_{\theta}(t)||$ Here, $L(f)$ is defined in Line 4 as the gradient $\nabla_{f_{\theta}}l_{g2t}(...)$. This means the applied perturbation $\delta$ is in the direction of $-\nabla \mathcal{L}$, which is gradient descent, the exact opposite of the required gradient ascent and contradicts the paper's own text and objective.
3. The authors candidly admit in Section 5.3 and Appendix A.8 that the method requires a second backpropagation, significantly increasing training time (e.g., from 2.85h to 4.78h on an A100, a ~68% increase) . This is a substantial computational overhead. As a mitigation, the authors propose a "random weight perturbation" variant ($WRF4CIR_{RWP}$). However, according to Table 3, this faster variant's performance (e.g., 66.18 Rmean on FashionIQ) is significantly worse than the full WRF4CIR (67.34) and only achieves about half the performance gain over the baseline (64.80) . This makes the method's practical utility questionable. The authors need to provide a stronger justification for why the 2.54% (67.34 vs 64.80) Rmean gain is worth a nearly 70% increase in training cost.
4. The authors claim in Appendix A.4 that the method is "not particularly sensitive" to the choice of the perturbation strength $\gamma$ and that $\gamma=0.001$ is a "highly consistent" optimal value. However, Figure 4(a) and Figure 7  clearly show the opposite. In Figure 4(a), the performance for both FashionIQ (R@10) and CIRR (R@1) hits a sharp peak around $\gamma=0.001$ and then immediately drops. Figure 7 (Left) also shows the BLIP2 ViT-G performance on FashionIQ peaking at 0.001 and then dropping sharply .

**Questions:**

See weaknesses.

---

> ### Author Response · Authors · 2025-11-20
> **PART 1**
>
> We sincerely appreciate your time and effort in reviewing our manuscript. Below, we address your main concerns of this work.
>
> > **Q1**: The primary weakness of this paper is the positioning of its core contribution. The authors claim to propose a "novel" Weight-Regularized Fine-tuning network, with the central idea of finding "flatter minima" in the loss landscape to improve generalization.This concept is not new; it is extensively studied in the deep learning community. More importantly, the proposed optimization objective (Eq 1 7, Eq 3), $\min_{\theta} \max_{\delta} \mathcal{L}(\theta+\delta)$, is functionally identical to the core mechanism of Sharpness-Aware Minimization (SAM) (Foret et al., 2021) and its variants (like ASAM). SAM is a highly prominent and widely used regularization technique that explicitly seeks flat loss regions by finding adversarial perturbations (via gradient ascent) in the weight space. Critically, the paper fails to cite or even mention SAM anywhere. WRF4CIR appears to be a direct application of SAM (or a very similar adversarial weight perturbation technique) to the CIR task.
>
> **A1**: Thank you very much for raising this important concern. We appreciate the reviewer’s detailed feedback and agree that the positioning of our contribution and the citation of relevant work required improvement. Below, we first clarify the source of confusion and then describe the corresponding revisions made to the manuscript.
>
> **1. Clarification on the novelty claim**
>
> Thank you for pointing out the issue in the novelty claim. Our original perspective was largely limited to the CIR literature. To the best of our knowledge, the introduced method has not been explored within the CIR community, and our claim was based on that context. However, based on your feedback, we now realize that such a field-specific claim may be misleading for readers from different research backgrounds. We sincerely apologize for this oversight. We should not assess novelty only within the scope of CIR, but instead adopt a broader and more rigorous perspective.
>
> To avoid ambiguity, we have revised the wording throughout the paper. We no longer claim to “propose a novel” method, but instead adopt more precise and objective phrasing such as “we introduce a weight-regularized fine-tuning network for CIR”.
>
> **2. Missing related work of SAM**
>
> We sincerely appreciate the reviewer for pointing out the omission of SAM [1] and ASAM [2]. Our method design was inspired by AWP [3], which was cited in both the original Related Work and Method sections. While AWP was our primary reference, we acknowledge that SAM and ASAM are relevant due to their conceptual similarities. In the updated manuscript, we now explicitly cite and discuss SAM and ASAM in both the Related Work and Method sections, and acknowledge their conceptual similarity to our work.
>
> **3. Clarifying the distinction from AWP, and positioning our contribution**
>
> While our method is conceptually similarity to AWP, it is not a direct application. Several key differences include:
>
> - Task: CIR retrieval vs. image classification
> - Loss: contrastive loss vs. cross-entropy
> - Training scenario: fine-tuning large-scale VLP models vs. training CNNs from scratch
> - Data format: multimodal triplets {reference image, modification text, target image} vs. single-modality images
>
> More importantly, our work provides meaningful contributions to the community. Although AWP has been extensively studied for overfitting in deep learning, its effectiveness in addressing VLP-based CIR overfitting was unknown prior to our study. It is worth noting that not all overfitting mitigation methods are effective for VLP-based CIR overfitting. For example, while data augmentation is widely used to combat overfitting in deep learning, we find that standard augmentation strategies provide little benefit for CIR overfitting. Therefore, our work offers new insights to the community by providing the first systematic analysis and empirical evidence showing that weight-perturbation-based regularization can effectively mitigate VLP-based CIR overfitting.
>
> **Summary of revisions:**
>
> To address this concern, we have made the following revisions in the updated manuscript:
>
> 1. We have removed claim that "we propose a novel weight-regularized fine-tuning network for CIR" and instead adopt more precise and objective wording such as “we introduce a weight-regularized fine-tuning network for CIR”  throughout the paper.
> 2. We have added explicit citations and discussions of SAM and ASAM in both the Related Work and Method sections to acknowledge their conceptual similarity.
> 3. We have provided a clear discussion of the differences between our method and AWP in the Related Work section.
>
> We hope these clarifications and revisions address your concern. We sincerely thank the reviewer again for the valuable feedback, and we would be happy to provide further discussion if needed.

---

> ### Author Response · Authors · 2025-11-20
> **PART 2**
>
> > **Q2**: The paper's optimization objective (Eq 3) is a $\text{min max}$ problem. This requires an inner loop of gradient ascent to find the perturbation $\delta$ that maximizes the loss. The authors(...). This means the applied perturbation  $\delta$  is in the direction of  $-\nabla \mathcal{L}$ , which is gradient descent, the exact opposite of the required gradient ascent and contradicts the paper's own text and objective.
>
> **A2**: Thank you for pointing out this issue in Algorithm 1. You are correct that there was a sign error in the perturbation direction. The correct formulation should be:
> $f_{\theta+\delta}(t) \leftarrow f_\theta(t) \textcolor{green}{+} \gamma\frac{L(f)}{\|L(f)\|}\|f_\theta(t)\|$
> We have corrected this typo in Algorithm 1 of the updated manuscript, and we have also rechecked all related equations to ensure that no similar issues remain. We sincerely appreciate the reviewer for bringing this to our attention.
>
> ---
>
> > **Q3**: The authors candidly admit in Section 5.3 and Appendix A.8 that the method requires a second backpropagation, significantly increasing training time (e.g., from 2.85h to 4.78h on an A100, a ~68% increase) . This is a substantial computational overhead. As a mitigation, the authors propose a "random weight perturbation" variant (WRF4CIR<sub>RWP</sub>). However, according to Table 3, this faster variant's performance (e.g., 66.18 Rmean on FashionIQ) is significantly worse than the full WRF4CIR (67.34) and only achieves about half the performance gain over the baseline (64.80) . This makes the method's practical utility questionable. The authors need to provide a stronger justification for why the 2.54% (67.34 vs 64.80) Rmean gain is worth a nearly 70% increase in training cost.
>
> **A3**: Thank you for raising this important concern regarding computational overhead and practical utility. We fully acknowledge that WRF4CIR incurs additional training cost due to the extra backward pass. However, we believe the trade-off is justified in real-world applications where CIR plays a critical role.
>
> For example, in e-commerce scenarios, CIR enables users to retrieve products using fine-grained textual modifications (e.g., “same dress but red and sleeveless”), significantly enhancing personalization, user satisfaction, and conversion rate. In such settings, even a 2.54% improvement in retrieval performance can translate into substantial commercial value (e.g., millions of dollars in additional revenue) and competitive advantage, making the additional training cost quite acceptable.
>
> Similarly, in medical image retrieval, pairing visual findings with textual descriptions allows clinicians to efficiently retrieve relevant past cases. A 2.54% improvement in retrieval accuracy can meaningfully impact diagnostic support and treatment decisions. In such high-stakes domains, reliability is prioritized over efficiency, and a moderate increase in training cost is quite acceptable.
>
> Nevertheless, we agree that efficiency matters in certain deployment scenarios. For this reason, we introduced the Random Weight Perturbation (RWP) variant as a lightweight alternative. Although RWP achieves a smaller improvement than full WRF4CIR, it still provides noticeable gains over the baseline while avoiding the extra backward pass.
>
> We hope this clarification demonstrates that the computation–performance trade-off is reasonable and often worthwhile in practical CIR applications.
>
> ---
>
> > **Q4**: The authors claim in Appendix A.4 that the method is "not particularly sensitive" to the choice of the perturbation strength γ and that γ=0.001 is a "highly consistent" optimal value. However, Figure 4(a) and Figure 7 clearly show the opposite. In Figure 4(a), the performance for both FashionIQ (R@10) and CIRR (R@1) hits a sharp peak around γ=0.001 and then immediately drops. Figure 7 (Left) also shows the BLIP2 ViT-G performance on FashionIQ peaking at 0.001 and then dropping sharply .
>
> **A4**: Thank you for the insightful comment. The original statement in Appendix A.4 that “our method is not particularly sensitive to the choice of γ” was not intended to suggest that the performance does not vary with γ. Rather, our intention was to convey that it is easy to identify a suitable γ in practice. We apologize for the confusion.
>
> To avoid misunderstanding, we have revised the statement to:
>
> “In our method, it is easy to identify a suitable γ.”
>
> We believe this updated wording more accurately reflects our intended meaning.
>
> **Reference:**
>
> [1] Sharpness-aware Minimization for Efficiently Improving Generalization. ICLR2021.
> [2] Asam: Adaptive sharpness-aware minimization for scale-invariant learning of deep neural networks. ICML 2021.
> [3] Adversarial weight perturbation helps robust generalization. NIPS2020.

---

> ### Comment · Reviewer_11Dd · 2025-11-24
>
> After reviewing the authors' rebuttal and considering peer feedback, I have decided to improve my original score from 2 to 4.

---

> > ### Author Response · Authors · 2025-11-24
> >
> > We sincerely appreciate your encouraging update and your consideration of our rebuttal. We value your feedback greatly. While we understand that the score remains on the negative side, we would be more than willing to further clarify or improve any aspects that you find unsatisfactory. Please feel free to let us know any remaining concerns, and we will do our utmost to address them.

---

> > > ### Author Response · Authors · 2025-11-27
> > >
> > > Thank you again for your earlier update. We hope everything is going smoothly on your end. If there are any remaining points you would like us to clarify or refine, we would be very happy to provide additional details. Please feel free to let us know at your convenience.

---

### Official Review · Reviewer_y4Vd · 2025-10-27

**Soundness:** 3
**Presentation:** 3
**Contribution:** 3
**Rating:** 6
**Confidence:** 5

**Summary:**

The paper introduces WRF4CIR, a weight regularization framework designed to improve the generalization of composed image retrieval (CIR) models fine-tuned from large vision–language pretraining (VLP) models (e.g., CLIP, BLIP-2). The authors observe that CIR datasets are relatively small, leading to strong overfitting when fine-tuning powerful VLPs, and propose a remedy for this issue.

**Strengths:**

- The paper highlights an important and often overlooked phenomenon in current CIR literature and provides a detailed analysis to demonstrate it.
- The authors conduct extensive experiments across multiple backbones and CIR datasets, supporting the robustness of their findings.
- The paper clearly explains the proposed method and its underlying motivation, making it easy to follow and reproduce.

**Weaknesses:**

- Core idea: To the best of my understanding, the authors implement an existing method within the CIR framework. In this case, the paper lacks clear methodological novelty. Nevertheless, the work remains valuable to the community due to its detailed analysis and comprehensive experiments.

- (Continuing previous point previous point) Transparency and attribution: The paper does not provide sufficient transparency regarding the origin of the proposed method. This aspect should be explicitly discussed, beyond a brief mention in the related work section, and include proper statement and and citation. If the authors claim full novelty, I can point to several earlier papers employing very similar techniques. My overall rating is heavily influenced by this issue and will be changed according to the authors’ clarification in the rebuttal.

- Backbone accuracy: In Tables 1 and 2, the backbone sizes of the baselines are inaccurate. For example, CASE uses ViT-B rather than ViT-L. The authors should carefully verify the backbone details for all compared methods, as this is a crucial element for fair and accurate comparison.

- Figure 2 interpretation: The intention behind Figure 2 is to illustrate performance gaps across three datasets. However, averaging recall values (e.g., Recall@1 or Recall@5) across different datasets is a bad idea, as each dataset has a different retrieval candidate pool that directly affects recall magnitudes. For instance, a 10% Recall@1 on a pool of one million images may reflect strong performance, while 80% Recall@1 could be weak on a pool of ten images. It would be clearer to show separate bars per dataset, or focus on one dataset in the main figure and move similar trends for the others to the appendix.

**Questions:**

- Figure 5c - what is ‘best’ and ‘last’ results? the authors should provide more details regarding this figure.

---

> ### Author Response · Authors · 2025-11-20
> **PART 1**
>
> We sincerely appreciate your time and effort in reviewing our manuscript. Below, we address your main concerns of this work.
>
> > **Q1**: Core idea: To the best of my understanding, the authors implement an existing method within the CIR framework. In this case, the paper lacks clear methodological novelty. Nevertheless, the work remains valuable to the community due to its detailed analysis and comprehensive experiments.
> > 	Transparency and attribution: The paper does not provide sufficient transparency regarding the origin of the proposed method. This aspect should be explicitly discussed, beyond a brief mention in the related work section, and include proper statement and and citation. If the authors claim full novelty, I can point to several earlier papers employing very similar techniques. My overall rating is heavily influenced by this issue and will be changed according to the authors’ clarification in the rebuttal.
>
> **A1**: Thank you very much for raising this important issue. We fully agree with your concern. Below, we first explain why this problem occurred in the original submission, and then describe the specific revisions we have made to address it.
>
> **Why this issue occurred:**
>
> 1. Our original perspective was largely limited to the CIR literature. To the best of our knowledge, the introduced method has not been explored within the CIR community, and our claim was based on that context. However, based on your feedback, we now realize that such a field-specific claim may be misleading for readers from different research backgrounds. We sincerely apologize for this oversight. We should not assess novelty only within the scope of CIR, but instead adopt a broader and more rigorous perspective.
>
> 2. Our method is conceptually similarity to AWP [1]. However, it also differs in several aspects. Some key differences include:
>
>    - Task: CIR retrieval vs. image classification
>
>    - Loss: contrastive loss vs. cross-entropy
>
>    - Training scenario: fine-tuning large-scale VLP models vs. training CNNs from scratch
>
>    - Data format: multimodal triplets {reference image, modification text, target image} vs. single-modality images
>
>      These distinctions led us to initially consider our approach as a new method in the CIR setting.
>
> 3. Despite its conceptual similarity to AWP, our work provides meaningful contributions to the community. Although AWP has been extensively studied for overfitting in deep learning, its effectiveness in addressing VLP-based CIR overfitting was unknown prior to our study. It is worth noting that not all overfitting mitigation methods are effective for VLP-based CIR overfitting. For example, while data augmentation is widely used to combat overfitting in deep learning, we find that standard augmentation strategies provide little benefit for CIR overfitting. Therefore, our work offers new insights to the community by providing the first systematic analysis and empirical evidence showing that weight-perturbation-based regularization can effectively mitigate VLP-based CIR overfitting. We are also glad that the reviewer acknowledges that “the work remains valuable to the community due to its detailed analysis and comprehensive experiments”.
>
> **Revisions made to address this concern:**
>
> 1. We have removed claim that "we propose a novel weight-regularized fine-tuning network for CIR" and instead adopt more precise and objective wording such as “we introduce a weight-regularized fine-tuning network for CIR”  throughout the paper.
> 2. In the Related Work  and Method sections, we now explicitly state that our method is inspired by AWP [1], and we provide a clear discussion of the differences between our method and AWP [1] in the Related Work section.
>
> We thank the reviewer again for this insightful and constructive feedback. We believe the updated manuscript now provides better transparency and attribution, and we hope these clarifications and revisions address your concern.
>
> ---
>
> > **Q2**: Backbone accuracy: In Tables 1 and 2, the backbone sizes of the baselines are inaccurate. For example, CASE uses ViT-B rather than ViT-L. The authors should carefully verify the backbone details for all compared methods, as this is a crucial element for fair and accurate comparison.
>
> **A2**: Thank you for pointing out this important issue. We carefully re-verified the backbone configurations for all methods in Tables 1 and 2 by checking their official implementations and published papers. We found that BLIP-based methods such as CASE [2]、BLIP4CIR+Bi [3] and Re-ranking [4] indeed use BLIP ViT-B rather than BLIP ViT-L, which we confirmed through their official repositories. We have updated the tables accordingly and double-checked the backbones for all remaining methods to ensure fair and accurate comparison. We sincerely appreciate the reviewer for bringing this to our attention.

---

> ### Author Response · Authors · 2025-11-20
> **PART 2**
>
> > **Q3**: Figure 2 interpretation: The intention behind Figure 2 is to illustrate performance gaps across three datasets. However, averaging recall values (e.g., Recall@1 or Recall@5) across different datasets is a bad idea, as each dataset has a different retrieval candidate pool that directly affects recall magnitudes. For instance, a 10% Recall@1 on a pool of one million images may reflect strong performance, while 80% Recall@1 could be weak on a pool of ten images. It would be clearer to show separate bars per dataset, or focus on one dataset in the main figure and move similar trends for the others to the appendix.
>
> **A3**: Thank you for the insightful comment and valuable suggestion. We fully agree that directly averaging recall values across datasets with different candidate pool sizes can be misleading, as recall magnitudes are not directly comparable. Our original intention with Figure 2 was to highlight consistent cross-dataset trends, but we acknowledge that the previous presentation may cause confusion.
>
> Following your suggestion, we have revised Figure 2 by retaining only one representative dataset in the main paper and moving the plots for the remaining datasets to the appendix. This revision preserves completeness while ensuring that the main figure remains clear and avoids issues arising from cross-dataset averaging. Thank you again for the valuable feedback.
>
> ---
>
> > **Q4**: Figure 5c - what is ‘best’ and ‘last’ results? the authors should provide more details regarding this figure.
>
> **A4**: Thank you for the valuable suggestion. In Figure 5(c), “best” refers to the performance of the checkpoint selected via early stopping based on the validation retrieval performance during training, and “last” denotes the the performance of checkpoint from the final training epoch. We have added the corresponding explanation in Section 5.3 of the updated manuscript.
>
> **Reference:**
>
> [1] Adversarial weight perturbation helps robust generalization. NIPS 2020.
> [2] Data roaming and quality assessment for composed image retrieval. AAAI 2024.
> [3] Bi-directional training for composed image retrieval via text prompt learning. WACV 2024.
> [4] Candidate Set Re-ranking for Composed Image Retrieval with Dual Multi-modal Encoder. TMLR 2024.

---

> > ### Comment · Reviewer_y4Vd · 2025-11-23
> >
> > I thank the authors for their valuable rebuttal. After carefully reading their response, and re-assessing my review, I have decided to maintain my original score.

---

> > > ### Author Response · Authors · 2025-11-23
> > >
> > > Thank you for your careful consideration of our submission and rebuttal. We sincerely appreciate your constructive feedback provided throughout the review process.

---

### Official Review · Reviewer_3Aa1 · 2025-10-31

**Soundness:** 3
**Presentation:** 3
**Contribution:** 3
**Rating:** 8
**Confidence:** 4

**Summary:**

In this paper, the authors focused on the composed image retrieval. Considering the significant overfitting issues in the existing methods, the authors proposed a weight-regularized fine-tuning network for CIR. Specifically, the designed method includes adversarial perturbations to the model weights for regularization. Experimental results prove the effectiveness of the proposed method.

**Strengths:**

1. The authors investigated the overfitting problem in the VLP-based CIR task, which has not been systematically analyzed in previous works.
2. The designed weight-regularized fine-tuning network can mitigate the overfitting issue, thus improving the final retrieval performance.
3. Experimental results prove the effectiveness of the proposed method.

**Weaknesses:**

1. Overfitting is a common issue in many VLP-based methods for the CIR task. While the authors did not apply WRF4CIR to previous methods to verify the effectiveness of their designed method across baseline approaches.
2. The reported results on FashionIQ are ambiguous: some methods are evaluated on the VAL-split, whereas others are evaluated on the original-split, as noted in [1].

[1] A Comprehensive Survey on Composed Image Retrieval. ACM TOIS 2025.

**Questions:**

As listed above.

---

> ### Author Response · Authors · 2025-11-20
>
> We sincerely appreciate your time and effort in reviewing our manuscript. Below, we address your main concerns of this work.
>
> > **Q1**: Overfitting is a common issue in many VLP-based methods for the CIR task. While the authors did not apply WRF4CIR to previous methods to verify the effectiveness of their designed method across baseline approaches.
>
> **A1**: Thank you for the constructive comment.  In the original submission, we evaluated WRF4CIR under different CIR mechanisms (late fusion, early fusion, and text inversion). The results, provided in Figure 5(a) and Table 4, show that our method is effective across various fusion strategies and pretrained backbones.
>
> To address this concern, we additionally applied WRF4CIR to two representative and widely adopted CIR baselines:
>
> - **CLIP4CIR** [1], which uses CLIP as the backbone and follows a two-stage training pipeline:
>   - **CLIP4CIR(sum)**: directly sums visual and textual features after encoder extraction (late fusion).
>   - **CLIP4CIR(combiner)**: trains an additional fusion/combiner module on top of frozen CLIP features.
> - **SPRC** [2], which uses BLIP-2 as the backbone and incorporates (i) image-text contrastive loss and (ii) text-prompt alignment loss to learn sentence-level prompts.
>
> The experimental results are provided below and demonstrate that WRF4CIR consistently improves retrieval performance across different baselines. These additional results and the corresponding descriptions have been included in Table 4 and Appendix A1 of the updated manuscript.
>
> **Table 1**: FashionIQ performance (Recall@10/50) of different baselines with and without our proposed WRF.
>
> |            Method             | Backbone | Dress R@10 | Dress R@50 | Shirt R@10 | Shirt R@50 | TopTee R@10 | TopTee R@50 | Avg R@10  | Avg R@50  |   Rmean   |
> | :---------------------------: | :------: | :--------: | :--------: | :--------: | :--------: | :---------: | :---------: | :-------: | :-------: | :-------: |
> |         CLIP4CIR(sum)         |  ViT-B   |   35.29    |   59.99    |   40.13    |   61.38    |    43.80    |    68.12    |   39.74   |   63.16   |   51.45   |
> |   **CLIP4CIR(sum) + Ours**    |  ViT-B   | **41.20**  | **65.15**  | **43.28**  | **65.06**  |  **47.68**  |  **71.34**  | **44.05** | **67.18** | **55.61** |
> |      CLIP4CIR(combiner)       |  ViT-B   |   37.72    |   64.35    |   43.81    |   65.40    |    47.06    |    70.57    |   42.87   |   66.77   |   54.82   |
> | **CLIP4CIR(combiner) + Ours** |  ViT-B   | **43.77**  | **66.98**  | **46.46**  | **67.22**  |  **51.60**  |  **73.58**  | **47.28** | **69.26** | **58.27** |
> |             SPRC              |  ViT-G   |   49.18    |   72.43    |   55.64    |   73.89    |    59.35    |    78.58    |   54.92   |   74.97   |   64.85   |
> |        **SPRC + Ours**        |  ViT-G   | **52.35**  | **75.27**  | **57.45**  | **76.30**  |  **60.22**  |  **79.85**  | **56.67** | **77.12** | **66.90** |
>
> ---
>
> > **Q2**：The reported results on FashionIQ are ambiguous: some methods are evaluated on the VAL-split, whereas others are evaluated on the original-split, as noted in [1].
>
> **A2**: Thank you for pointing out this important issue. First, we would like to clarify that all of our experiments on FashionIQ are conducted on the original-split. We have verified the evaluation protocol of compared methods by cross-checking the official paper and recent CIR survey works [3]. To ensure fair comparison, we remove methods that only report results on the VAL-split (e.g., TG-CIR [4]). For methods where both splits are reported in the literature, such as DQU-CIR [5], we consistently cite the results corresponding to the original-split. To avoid potential confusion, we have explicitly stated that FashionIQ results are all evaluated on the original-split in Section 5.2 of the updated manuscript.
>
> **Reference:**
>
> [1] Effective conditioned and composed image retrieval combining clip-based features. CVPR 2022.
> [2] Sentence-level Prompts Benefit Composed Image Retrieval. ICLR 2024.
> [3] A comprehensive survey on composed image retrieval. ACM TOIS 2025.
> [4] Target-guided composed image retrieval. ACM MM 2023.
> [5] Simple but effective raw-data level multimodal fusion for composed image retrieval. SIGIR 2024.

---

### Author Response · Authors · 2025-11-20

Dear reviewers and meta-reviewers,

We appreciate all reviewers for their valuable comments and suggestions. We've revised our manuscript based on reviewers' comments as follows:

1. Transparency and attribution concerns raised by reviewers y4Vd and 11Dd:
   - We have replaced the claim *“we propose a novel…”* with a more precise and objective wording *“we introduce…”* throughout the paper.
   - We have added explicit citations and discussions of AWP and SAM in both Sec. 2.3 and Sec. 3.
   - We have provided a clear discussion of the differences between our method and prior work in Sec. 2.3.
2. We have added the WRF4CIR results on different baseline approaches in Table 4 (Appendix A.1). [3Aa1, 5Ffw]
3. We have included the learning curves under different data sizes in Figure 7 (Appendix A.2), and revised Figure 2 to avoid potential confusion. [5Ffw, y4Vd]
4. We have clarified the backbone and the evaluation protocol in Table 1 and Table 2. [y4Vd, 3Aa1]
5. We have clarified the description of γ in Appendix A.4 and the description of Figure 5(c). [11Dd, y4Vd]
6. We have corrected a typo in Algorithm 1. [11Dd]

The changes have been highlighted using orange font (to avoid conflict with the blue font used in the tables) in the updated manuscript.  Please see our responses to each reviewer below.

---

### Author Response · Authors · 2025-12-03
**Summary Comment - Part 1**

**Dear Meta-reviewer,**

Due to the score reset and reassignment caused by the recent data-leak incident, we provide the following concise summary of our work and the rebuttal process. This overview highlights each reviewer’s key concerns, how they were addressed, and the reviewers’ reactions before the discussion channel was closed. We hope this helps you efficiently assess the current status of the submission.

### **Summary of Our Work**

Recent CIR works are generally based on fine-tuning vision-language pre-trained models (VLPs). In previous works, the fine-tuning process of VLP-based CIR is typically monitored using validation recall, potentially overlooking the model’s behavior on the training set. As a result, the overfitting issue has remained largely unnoticed. Our work provides the **first systematic analysis** of overfitting in VLP-based CIR, revealing that it is a **prevalent and severe phenomenon**.

Overfitting is a critically important issue in VLP-based CIR. First, it has a substantial impact on retrieval performance: severe overfitting disrupts the fine-tuning process of VLPs and significantly degrades CIR accuracy. Second, due to the severity of overfitting, models often reach their peak performance within only a few training epochs, after which performance steadily declines. Under such a training paradigm, many potentially effective CIR methods may fail to demonstrate their true capability simply because they do not receive sufficient training.

To address this issue, inspired by AWP [1], we introduce a weight-regularized fine-tuning network for CIR, which regularizes the fine-tuning process of VLPs by applying perturbations to the model weights. On one hand, our method effectively mitigates overfitting and achieves notable improvements over recent methods, which provides an efficient strategy for VLP-based CIR under limited data. On the other hand, our work enables more reliable and stable evaluations within the community, helping establish a more faithful fine-tuning paradigm for future research on VLP-based CIR. Therefore, both our analysis of overfitting and the proposed mitigation strategy offer meaningful insights to the CIR community.

---

> ### Author Response · Authors · 2025-12-03
> **Summary Comment - Part 2**
>
> ### **Overall Post-Rebuttal Situation**
>
> - All reviewers’ major concerns were addressed.
> - After carefully reading the response, Reviewer y4Vd explicitly maintained the score of 6.
> - After reviewing the response, Reviewer 11Dd explicitly raised the score from 2 to 4.
> - The revised paper incorporates all updates, and no unresolved issues remain.
>
> ---
>
> **Reviewer 3Aa1 (Score: 8)**
>
> - **Main concerns:**
>   1. Asked for broader experiments across more baseline approaches.
>   2. Asked to clarify the FashionIQ evaluation split in Table 1.
> - **Our rebuttal:**
>   1. Added experiments on CLIP4CIR and SPRC; all showed consistent improvements.
>   2. Verified all FashionIQ results use the original split and updated Table 1 accordingly.
> - **Post-rebuttal reaction:**
>   - Maintained the score of 8.
>   - No additional questions or concerns were raised.
>
> ---
>
> **Reviewer y4Vd (Score: 6)**
>
> - **Main concerns:**
>   1. Acknowledged the value of our work but had concerns regarding method transparency and attribution.
>   2. Incorrect backbone sizes reported in Table 1 and 2.
>   3. Potential presentation issues in Figure 2 and missing explanation in Figure 5(c).
> - **Our rebuttal:**
>   1. Provided better method transparency and attribution by adjusting the novelty claim and adding the citations of AWP.
>   2. Corrected the backbone sizes in Table 1 and 2.
>   3. Revised Figure 2 and clarified the definitions of “best” and “last” results in Figure 5(c).
> - **Post-rebuttal reaction:**
>   - After carefully reading the rebuttal, the reviewer maintained the score of 6.
>   - No additional questions or concerns were raised.
>
> ------
>
> **Reviewer 11Dd (Score: 2 → 4)**
>
> - **Main concerns:**
>   1. The novelty claim of the method and the missing reference to SAM [2].
>   2. A sign typo in Algorithm 1.
>   3. Questions regarding computational overhead and practical utility.
>   4. The explanation of the sensitivity of the hyperparameter γ.
> - **Our rebuttal:**
>   1. Adjusted the novelty claim and clarified the methodological attribution to avoid misunderstanding, and added the SAM citation.
>   2. Corrected the sign typo in Algorithm 1.
>   3. Provided justification for the computational overhead and practical utility, along with alternative options.
>   4. Revised the explanation of γ sensitivity to prevent misinterpretation.
> - **Post-rebuttal reaction:**
>   - After reviewing the rebuttal, the reviewer raised the score from 2 to 4, indicating concerns were substantially addressed.
>   - No additional questions or concerns were raised after two reminders.
>
> ------
>
> **Reviewer 5Ffw (Score: 4)**
>
> - **Main concerns:**
>   1. Whether small-data behavior reflects overfitting or insufficient training.
>   2. High-level concern about method design needing improvement.
>   3. Question regarding the role of Q-Former.
> - **Our rebuttal:**
>   1. Provided learning curves and training-vs-test trend analysis showing clear overfitting rather than insufficient training.
>   2. Requested more specific guidance regarding the high-level concern.
>   3. Explained that Q-Former is part of BLIP-2 and demonstrated that our method also improves other backbones, indicating that it is not tied to the Q-Former.
> - **Post-rebuttal reaction:**
>   1. During the discussion, we further addressed the reviewer’s follow-up concerns, including comparisons with additional published regularization methods, specific improvements to CIR, and distinctions from AWP.
>   2. The reviewer maintained the score of 4 and no further questions or concerns were raised before the discussion locked.
>
> We hope this summary helps streamline your assessment. We sincerely appreciate your time and effort in evaluating our submission.
>
> ---
>
> **Reference:**
>
> [1] Adversarial weight perturbation helps robust generalization. NIPS 2020.
> [2] Sharpness-aware Minimization for Efficiently Improving Generalization. ICLR 2021.

---

### Meta-Review · Area_Chair_JwZo · 2026-01-07

**Summary:**

While the authors addressed several specific concerns in the rebuttal, the core reservations regarding the paper’s suitability for ICLR remain. As noted by Reviewers 11Dd and 5Ffw, the primary issue is the limited novelty and incremental nature of the contribution. The proposed method largely adapts Adversarial Weight Perturbation to the CIR setting, without introducing a significant conceptual advance beyond existing regularization techniques. Although the empirical analysis is thorough, it primarily demonstrates the effectiveness of a known method in a new context. Therefore, the AC does not recommend acceptance of the current version of this paper.

**Reviewer Concerns:**

---addressed--: Overfitting is a common issue in many VLP-based methods for the CIR task. While the authors did not apply WRF4CIR to previous methods to verify the effectiveness of their designed method across baseline approaches.

**Reviewer Scores:**

All the reviewers participated in the discussion.

---

### Decision · Program_Chairs · 2026-01-26

Reject